# Blending Complementary Memory Systems in Hybrid Quadratic-Linear Transformers

**Kazuki Irie**[1]   **Morris Yau**[2]   **Samuel J. Gershman**[1,3]
[1]Department of Psychology and Center for Brain Science,
Harvard University, Cambridge, MA, USA
[2]MIT CSAIL, Cambridge, MA, USA
[3]Kempner Institute for the Study of Natural and Artificial Intelligence,
Harvard University, Cambridge, MA, USA
{kirie, gershman}@fas.harvard.edu, morrisy@mit.edu

## Abstract

We develop hybrid memory architectures for general-purpose sequence processing neural networks, that combine *key-value memory* using softmax attention (KV-memory) with *fast weight memory* through dynamic synaptic modulation (FW-memory)—the core principles of quadratic and linear transformers, respectively. These two memory systems have complementary but individually limited properties: KV-memory offers precise retrieval but is constrained by quadratic complexity in sequence length, while FW-memory supports arbitrarily long sequences and enables more expressive computation but sacrifices precise recall. We propose and compare three methods to blend these two systems into a single memory system, differing in how and when input information is delivered to each system, to leverage the strengths of both. We conduct experiments on general language modeling and retrieval tasks by training 340M- and 1.3B-parameter models from scratch, as well as on synthetic algorithmic tasks designed to precisely illustrate the benefits of certain hybrid methods over others. We also evaluate our hybrid memory systems on reinforcement learning in partially observable environments. Overall, we demonstrate how a well-designed hybrid can overcome the limitations of its individual components, offering new insights into the design principle of neural memory systems.[1]

## 1 Introduction

Modern transformers come in two flavors. The standard or "quadratic transformer" (QT) [1] leverages softmax attention over explicit key-value memory storage which enables precise memory retrieval [2], while the length of sequences they can handle is limited in practice due to its quadratic computational complexities (as reflected in our model terminology). On the other hand, "linear transformers" (LT), also known as fast weight programmers (FWPs) [3, 4] sacrifice the precision of the softmax function (the complexity bottleneck of QT), but in return they support the processing of arbitrarily long sequences. Furthermore, some of their variants (called DeltaNet models [5–8]) enable more expressive computations (e.g., certain types of state-tracking), which QTs struggle to perform [7–16].

While the unique advantages of quadratic and linear transformers may seem inherently incompatible to achieve simultaneously, they are all desirable in an ideal general-purpose memory system. How could we achieve multiple computational properties that are incompatible within a single system? The brain's apparent solution is to integrate multiple types of memory mechanisms; each with distinct properties that meet specific demands under various constraints, thereby enabling flexible problem

---

[1]Our code is public: https://github.com/kazuki-irie/hybrid-memory.

39th Conference on Neural Information Processing Systems (NeurIPS 2025).

Table 1: Complementarity of the memory systems in two transformer types.

| Property | Key-value memory | Fast weight memory |
|---|---|---|
| Complexity | quadratic | **linear** |
| Context length | bounded | **unbounded** |
| Retrieval precision | **high** | low |
| Expressivity | low | **high** |

solving. One example of such a view is the classic Complementary Learning Systems architecture [17, 18], which hypothesizes that the brain has a division of labor into separate (but interacting) episodic and semantic memory systems whose functions are incompatible (remembering specifics vs. extracting generalities). Here we propose a *different* division of labor into complementary linear and quadratic memory systems each with complementary strengths (Table 1).

We propose and compare three blending methods that are conceptually well-motivated—they differ in how and when input information is delivered to each memory system, and empirically evaluate them to ultimately converge on a single architecture. We conduct experiments to test general language modeling and in-context retrieval abilities (using the standard `lm-evaluation-harness` [19]), by training 340M- and 1.3B-parameter language models from scratch using 15B tokens of the HuggingFace FineWeb-Edu dataset [20]. We also conduct experiments on synthetic algorithmic tasks (parity and modular arithmetics) [7] to determine the type of hybrid methods that can preserve the expressivity advantages of DeltaNet (which we use as the LT/FWP component of our hybrid)—a critical discussion and evaluation that are completely absent in prior discussions on quadratic and linear transformer hybrids [21, 22]. Overall, we demonstrate the benefits of hybrid model designs that can take the full advantage of the latest LT/FWP models, and show how a well-designed hybrid can mitigate the limitations of its individual components; this offers novel insights into the design principle of neural memory systems.

Finally, all our models are scalable as they are compatible with the algorithms and implementations for highly efficient quadratic and linear transformers that have been proposed in prior work on `flash-attention` [23] and `flash-linear-attention` [24] (see our code link on page 1).

## 2 Background

Here we provide background necessary to describe the proposed hybrid models in Sec. 3, namely: key-value memory (Sec. 2.1) and fast weight programming/memory (Sec. 2.2), which are the cornerstones of quadratic and linear transformers, respectively.

**Common Settings.** Let $d_{\text{in}}, d_{\text{out}}, d_{\text{key}}, t$ denote positive integers. All the models studied here are causal sequence processing neural networks, which, at every time step $t$, receive an input $\boldsymbol{x}_t \in \mathbb{R}^{d_{\text{in}}}$ and produce an output $\boldsymbol{y}_t \in \mathbb{R}^{d_{\text{out}}}$ while maintaining an internal memory of all inputs received so far.

### 2.1 Key-value memory & Softmax attention

The main sequence processing computation in the standard/quadratic transformer [1] is the self-attention operation, which, at every time step $t$, receives an input $\boldsymbol{x}_t \in \mathbb{R}^{d_{\text{in}}}$ and produces an output $\boldsymbol{y}_t \in \mathbb{R}^{d_{\text{out}}}$, while maintaining the *key-value memory*, represented as two matrices $\boldsymbol{K}_t \in \mathbb{R}^{d_{\text{key}} \times t}$ and $\boldsymbol{V}_t \in \mathbb{R}^{d_{\text{out}} \times t}$ as follows:

$$[\boldsymbol{q}_t, \boldsymbol{k}_t, \boldsymbol{v}_t] = \boldsymbol{W}^{\text{slow}} \boldsymbol{x}_t \tag{1}$$

$$\boldsymbol{K}_t = \boldsymbol{K}_{t-1} \oplus \boldsymbol{k}_t \,;\, \boldsymbol{V}_t = \boldsymbol{V}_{t-1} \oplus \boldsymbol{v}_t \tag{2}$$

$$\boldsymbol{y}_t = \boldsymbol{V}_t \text{softmax}(\boldsymbol{K}_t^\top \boldsymbol{q}_t) \tag{3}$$

where $\boldsymbol{q}_t, \boldsymbol{k}_t \in \mathbb{R}^{d_{\text{key}}}, \boldsymbol{v}_t \in \mathbb{R}^{d_{\text{out}}}$ (with $[]$ denoting vector concatenation), $\boldsymbol{W}^{\text{slow}} \in \mathbb{R}^{(2d_{\text{key}}+d_{\text{out}}) \times d_{\text{in}}}$ is the trainable weight matrix (meaning of the superscript "slow" becomes clear in Sec. 2.2), $\oplus$ as in $\boldsymbol{K}_{t-1} \oplus \boldsymbol{k}_t$ denotes concatenation of vector $\boldsymbol{k}_t \in \mathbb{R}^{d_{\text{key}}}$ to matrix $\boldsymbol{K}_{t-1} \in \mathbb{R}^{d_{\text{key}} \times (t-1)}$ along the time dimension, yielding $\boldsymbol{K}_t \in \mathbb{R}^{d_{\text{key}} \times t}$ ($\boldsymbol{K}_0$ and $\boldsymbol{V}_0$ are initially empty). We omit the $1/\sqrt{d_{\text{key}}}$ scaling inside $\text{softmax}$, as well as the output projection, which are irrelevant to our discussion.

Eq. 3 is the so-called softmax attention operation; the query vector from the current step $t$ is compared to the keys from all the time steps through dot product; these similarity scores are sharpened through the softmax function (crucial for precise retrieval), and the resulting scores are used as coefficients to compute the weighted average of all the value vectors to produce the output. As can be seen in Eqs. 2, the sizes of key and value memory storage linearly grow with the time step (i.e., sequence length), resulting in quadratic complexity w.r.t. sequence length in the attention computation in Eq. 3. Consequently, practical self-attention requires predetermining a maximum context length, i.e., size of the sliding window, and any input that falls outside the window is discarded. On the other hand, training of such a model can be highly efficient on modern hardware as the computation above is parallelizable over time steps/sequence elements.

## 2.2 Fast weight memory & Linear attention and delta rule

By replacing the softmax normalization in Eq. 3 by a non-linear activation function applied to each key and query vectors individually (which we denote $\phi$), the attention computation of Eqs. 2-3 can be reorganized [3, 25, 5, 26] to derive the so-called "recurrent form" of linear attention (LA) or fast weight programming (FWP) operation. Let $\otimes$ denote outer product. The "recurrent form" of a linear attention [3, 4] is a sequence operation that, at each time step $t$, transforms an input $\boldsymbol{x}_t \in \mathbb{R}^{d_{\text{in}}}$ to an output $\boldsymbol{y}_t \in \mathbb{R}^{d_{\text{out}}}$, while maintaining the so-called fast weight matrix $\boldsymbol{W}_t \in \mathbb{R}^{d_{\text{out}} \times d_{\text{key}}}$ as a memory state, as follows:

$$[\boldsymbol{q}_t, \boldsymbol{k}_t, \boldsymbol{v}_t] = \boldsymbol{W}^{\text{slow}}\boldsymbol{x}_t \tag{1}$$

$$\boldsymbol{W}_t = \boldsymbol{W}_{t-1} + \boldsymbol{v}_t \otimes \phi(\boldsymbol{k}_t) \tag{4}$$

$$\boldsymbol{y}_t = \boldsymbol{W}_t \phi(\boldsymbol{q}_t) \tag{5}$$

where Eq. 1 remains the same as in Sec. 2.1, and the "fast-changing" weight matrix $\boldsymbol{W}_t \in \mathbb{R}^{d_{\text{out}} \times d_{\text{key}}}$ is initially set to 0, i.e., $\boldsymbol{W}_0 = 0$. This can be viewed as a system of two networks where one net (the slow net; Eq. 1) learns to "program" the fast net (Eq. 5) by generating its weight changes (Eq. 4)—the origin of their names. Given that the computation per step is constant w.r.t. the time step/sequence length, the time complexity of this model is linear w.r.t. sequence length.

Note that the recurrent form of LA derived by Katharopoulos et al. [3] introduces an extra temporal variable ($\boldsymbol{z}_t = \boldsymbol{z}_{t-1} + \phi(\boldsymbol{k}_t)$; with $\boldsymbol{z}_0 = 0$) which is used to compute the normalizing term ($\boldsymbol{z}_t^\top \phi(\boldsymbol{q}_t)$) applied to the output (Eq. 5). However, more recent work has shown that such extra computation is unnecessary in practice [5, 27–29] with an appropriate choice of $\phi$ (discussed below).

We refer to the constant-size, context-dependent, time-varying fast weight matrix $\boldsymbol{W}_t$ as *fast weight memory* (a terminology which has also been used in past work augmenting an LSTM with other fast weight programming operations [30]), to contrast it with the *key-value memory* of Sec. 2.1. We denote them as FW-memory and KV-memory, respectively, for short.

**Chunk-wise Parallel Training.** In practice, using the linear-complexity recurrent form of LA above for training is inefficient as it is a purely sequential computation. Instead, a more practical method for training is a "hybrid" approach, called chunk-wise parallel training. The main idea is to divide a training sequence, as well as the corresponding computation, into small chunks; and causally process one chunk after another. The intra-chunk computation leverages the parallel computation, while the inter-chunk contribution uses the recurrent form. More formally, let $S$ and $\mathbf{n}$ denote positive integers; all the outputs in the $\mathbf{n}$-th chunk of size $S$, denoted as $\mathbf{Y_n} \in \mathbb{R}^{d_{\text{out}} \times S}$, can be computed as:

$$\mathbf{Y_n} = \mathbf{W_n Q_n} + \mathbf{V_n}(\mathbf{K_n^\top Q_n} \odot \mathbf{M}) \ ; \ \ \mathbf{W_{n+1}} = \mathbf{W_n} + \mathbf{V_n K_n^\top} \tag{6}$$

where $\mathbf{Q_n}, \mathbf{K_n} \in \mathbb{R}^{d_{\text{key}} \times S}$ and $\mathbf{V_n} \in \mathbb{R}^{d_{\text{out}} \times S}$ denote the matrices containing the query, key, value vectors within chunk $\mathbf{n}$, and $\mathbf{W_n} \in \mathbb{R}^{d_{\text{out}} \times d_{\text{key}}}$ is the fast weight memory up to chunk $\mathbf{n}$ (exclusive), with $\mathbf{W_0} = 0$, $\mathbf{M} \in \mathbb{R}^{S \times S}$ is the causal mask, and $\odot$ denotes element-wise multiplication. The first and second terms of Eq. 6 (left) correspond to the inter- and intra-chunk computations, respectively (while Eq. 6 (right) is the fast weight memory update). This results in an efficient sub-quadratic complexity algorithm for training LA (and other FWPs) in practice [31, 27, 28].

**Delta rule extension.** Motivated by prior practices and successes of replacing purely additive Hebbian weight modification rule by the delta rule [32, 33], DeltaNet [5] replaces the purely sum

update rule of LA (Eq. 4) by an update rule with a functional form of the classic delta rule [34]:

$$[\boldsymbol{q}_t, \boldsymbol{k}_t, \boldsymbol{v}_t, \beta_t] = \boldsymbol{W}_{\text{slow}} \boldsymbol{x}_t \tag{7}$$

$$\boldsymbol{W}_t = \boldsymbol{W}_{t-1} + \sigma(\beta_t)(\boldsymbol{v}_t - \boldsymbol{W}_{t-1}\phi(\boldsymbol{k}_t)) \otimes \phi(\boldsymbol{k}_t) \tag{8}$$

$$\boldsymbol{y}_t = \boldsymbol{W}_t \phi(\boldsymbol{q}_t) \tag{5}$$

where the dimension of $\boldsymbol{W}^{\text{slow}}$ is increased to be $\in \mathbb{R}^{(2*d_{\text{key}}+d_{\text{out}}+1)\times d_{\text{in}}}$, i.e., one extra output dimension ($+1$) is introduced to produce an addition variable $\beta_t \in \mathbb{R}$. $\sigma$ is an activation function applied to $\beta_t$, which is typically sigmoid or 2 times sigmoid (the factor 2 introduces negative eigenvalues in the state transition matrix enabling strong state-tracking abilities [7]). The choice of $\phi$ is particularly important for stability when the delta rule is used [5]; here we define $\phi$ as element-wise sigmoid linear unit (SiLU; $x$ times sigmoid) followed by $L_2$ normalization as in Yang et al. [29]. Eq. 8 corresponds to a rank-one update of the fast weight matrix, from $\boldsymbol{W}_{t-1}$ to $\boldsymbol{W}_t$, through the delta learning rule [34], where the slow net-generated variables, $\boldsymbol{v}_t, \phi(\boldsymbol{k}_t)$, and $\sigma(\beta_t)$, play the role of *target*, *input*, and *learning rate* of the delta rule, respectively.

DeltaNet has seen a recent revival as Yang et al. [29] developed a scalable chunk-wise parallel training algorithm for it. While prior extensions of DeltaNet have solely focused on improving its expressivity [35, 36, 10], recent extensions remarkably preserve the efficient parallel training property [6, 8]. DeltaNet distinguishes itself from other linear transformers due to its capabilities for state tracking [7, 8]; therefore, we'll adopt DeltaNet as the FW-memory component in our hybrid models. Nevertheless, our hybrid approach can naturally extend to other LT models or DeltaNet extensions.

## 3 Method: Hybrid Quadratic-Linear Transformers

We study three strategies for hybridizing FW-memory (Sec. 2.2) and KV-memory (Sec. 2.1) in a single system: *Delayed-Streaming* (Sec. 3.1), *Delayed-Chunk* (Sec. 3.2), and *Synchronous* (Sec. 3.3) approaches. As we'll show, each of these three approaches is motivated by different arguments and has connections to prior work. We conceptually discuss their advantages and validate them empirically in Sec. 4. We refer to these Hybrids of Quadratic and Linear Transformers as HQLTs.

### 3.1 Delayed-Streaming HQLT

The first variation of HQLT we discuss here is derived by the main goal of addressing the context window size limitation of KV-memory (Sec. 2.1). A typical KV-memory operates on a sliding window with a limited size covering only recent past, while discarding key-value pairs which fall outside of such KV-memory span. In the proposed "Delayed-Streaming HQLT", instead of completely discarding old key-value pairs, they are integrated into a separate FW-memory. This yields the following sequence model.

Let $\ominus$ denote the 'remove' operation taking a matrix and one of its column vectors to be removed as arguments. $S$ denotes the window size, i.e., the number of key-value pairs KV-memory can store.

At every time step $t$, an HQLT takes an input $\boldsymbol{x}_t \in \mathbb{R}^{d_{\text{in}}}$, and produces an output $\boldsymbol{y}_t \in \mathbb{R}^{d_{\text{out}}}$, while maintaining two types of memory: (1) KV-memory, denoted as two matrices $\boldsymbol{K}_{t-1} \in \mathbb{R}^{d_{\text{key}} \times S}$ and $\boldsymbol{V}_{t-1} \in \mathbb{R}^{d_{\text{out}} \times S}$, and (2) FW-memory, denoted as a matrix $\boldsymbol{W}_{t-1} \in \mathbb{R}^{d_{\text{out}} \times d_{\text{key}}}$. As in DeltaNet (Sec. 2.2), the trainable parameter matrix is $\boldsymbol{W}^{\text{slow}} \in \mathbb{R}^{(2*d_{\text{key}}+d_{\text{out}}+1)\times d_{\text{in}}}$, and variables are first generated from the input as follows:

$$[\boldsymbol{q}_t, \boldsymbol{k}_t, \boldsymbol{v}_t, \beta_t] = \boldsymbol{W}^{\text{slow}} \boldsymbol{x}_t \tag{9}$$

KV-memory is updated by removing the oldest key/value pair $(\boldsymbol{k}_{t-S}, \boldsymbol{v}_{t-S})$ from the memory, and adding the newly generated one $(\boldsymbol{k}_t, \boldsymbol{v}_t)$:

$$\boldsymbol{K}_t = \boldsymbol{K}_{t-1} \ominus \boldsymbol{k}_{t-S} \oplus \boldsymbol{k}_t \tag{10}$$

$$\boldsymbol{V}_t = \boldsymbol{V}_{t-1} \ominus \boldsymbol{v}_{t-S} \oplus \boldsymbol{v}_t \tag{11}$$

In Delayed-Streaming HQLT, the key/value pair removed from KV-memory is fed to FW-memory (Sychronous HQLT differs in this aspect; as we'll see in Sec. 3.3):

$$\boldsymbol{W}_t = \text{UpdateRule}(\boldsymbol{W}_{t-1}, \boldsymbol{k}_{t-S}, \boldsymbol{v}_{t-S}, \beta_t) \tag{12}$$

$$= \boldsymbol{W}_{t-1} + \sigma(\beta_t)(\boldsymbol{v}_{t-S} - \boldsymbol{W}_{t-1}\phi(\boldsymbol{k}_{t-S})) \otimes \phi(\boldsymbol{k}_{t-S}) \tag{13}$$

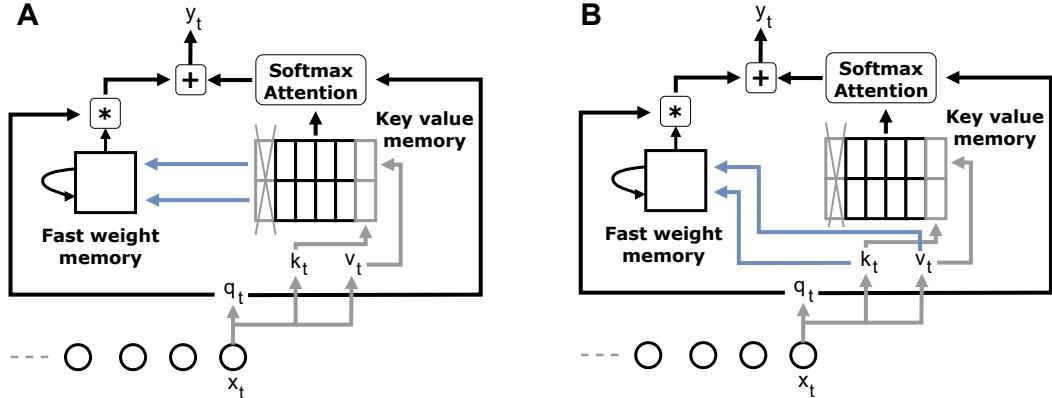

Figure 1: An illustration for Hybrid Quadratic-Linear Transformers (HQLTs). Two variations are shown. In the "Delayed-Stream" variant (**A**), the newly generated key/value pair is only fed to the key-value memory (KV-memory) system, and the old key/value pair that falls outside the context window of KV-memory is fed to the fast weight memory (FW-memory) system. In the "Synchronous" variant (**B**), the key/value pair generated at time step $t$ is fed to both KV-memory and FW-memory systems. The dynamic learning rate variable $\beta_t$ and memory mixing variable $\gamma_t$ are omitted.

This last equation corresponds to the delta rule as in Eq. 8. Note, however, that the choice of UpdateRule is arbitrary in principle; e.g., Gated Delta Rule [6] or Delta Product Rule [8] could be used. Here we focus on the most basic DeltaNet.

Finally, the output is computed by combining outputs from the two memory systems:

$$\boldsymbol{y}_t = \text{FastWeightMemory}(\boldsymbol{W}_t, \boldsymbol{q}_t) + \text{KeyValueMemory}(\boldsymbol{K}_t, \boldsymbol{V}_t, \boldsymbol{q}_t) \tag{14}$$

$$= \boldsymbol{W}_t \phi(\boldsymbol{q}_t) + \boldsymbol{V}_t \text{softmax}(\boldsymbol{K}_t^{\mathsf{T}} \boldsymbol{q}_t) \tag{15}$$

In this Delayed-Streaming HQLT, the division of labors between the two memory systems is as follows: FW-memory $\boldsymbol{W}_t$ is responsible for storing all the relevant information for all time steps $\tau \leq t - S$ (anything that are older than $S$ time steps), while KV-memory $(\boldsymbol{K}_t, \boldsymbol{V}_t)$ enables precise retrieval on the $S$ most recent context $t - S < \tau \leq t$.

HQLT achieves this untruncated-context sequence modeling with a fixed memory size corresponding to the total of KV and FW memory matrices, i.e., $S(d_{\text{in}} + d_{\text{out}}) + d_{\text{out}} * d_{\text{in}}$. The model illustration can be found in Figure 1A.

**Memory mixing/gating.** We further enhance Eq. 15 by exploring weighted mixing strategies between the FW-memory and KV-memory outputs, by generating additional context-dependent weights in Eq. 9 by increasing the output dimension of $\boldsymbol{W}^{\text{slow}}$ accordingly. We refer to Eq. 15 as *sum mixing*. In addition, we propose *dynamic scalar mixing* which generates two scalar variables $\alpha_t^{\text{FW}}, \alpha_t^{\text{KV}} \in [0, 1]$ (we apply sigmoid to achieve this bound) as:

$$\boldsymbol{y}_t = \alpha_t^{\text{FW}} * \text{FastWeightMemory}(\boldsymbol{W}_t, \boldsymbol{q}_t) + \alpha_t^{\text{KV}} * \text{KeyValueMemory}(\boldsymbol{K}_t, \boldsymbol{V}_t, \boldsymbol{q}_t) \tag{16}$$

and *dynamic vector mixing* by generating a vector variable $\gamma_t \in \mathbb{R}^{d_{\text{out}}}$ which is used as:

$$\boldsymbol{y}_t = \gamma_t \odot \text{FastWeightMemory}(\boldsymbol{W}_t, \boldsymbol{q}_t) + (1 - \gamma_t) \odot \text{KeyValueMemory}(\boldsymbol{K}_t, \boldsymbol{V}_t, \boldsymbol{q}_t) \tag{17}$$

The dynamic scalar mixing variation is akin to the one used in prior work [22], except that we generate and apply a separate scalar on each term.

### 3.2 Delayed-Chunk HQLT

While Delayed-Streaming HQLT above seems natural from the perspective of streaming causal sequence processing, another "delayed" variation can be motivated by the chunk-wise training algorithm of Eq. 6 (left). In fact, an HQLT can be naturally obtained by introducing the softmax

activation in the intra-chunk attention operation of Eq. 6 (left; the second term), i.e., $(\mathbf{K}_\mathbf{n}^\top \mathbf{Q}_\mathbf{n} \odot \mathbf{M})$; and by using this chunk-wise processing for inference too. We call this approach "Delayed-Chunk HQLT". We provide further comments about this model in Appendix B.1.

This model also directly connects to a closely related prior model proposed in Munkhdalai et al. [22] (see also [37]). However, we'll see how these "delayed" architectures are sub-optimal compared to the synchronous approach below, in light of recent advances in DeltaNet models, and to design hybrid models capable of leveraging the expressivity advantage of their FW-memory component.

### 3.3 Synchronous HQLT

The delayed versions are conceptually elegant in their approach to introducing a division of labor based on the age of key-value pairs in memory (KV-memory is responsible for recent ones, while FW-memory takes care of old ones). However, this is not compatible with the idea of potentially leveraging the expressivity complementarity: recent advances in FWPs demonstrate an expressivity advantage of DeltaNet over softmax attention [7, 8, 10]. Thus, in a hybrid system, FW-memory could play a crucial role when the system has to deal with types of computations that KV-memory cannot perform, and this may require FW-memory to also process the most recent inputs.

This motivates "Synchronous HQLT" in which both KV-memory and FW-memory process the same input simultaneously. The corresponding equations are identical to those of Delayed-Streaming HQLT (Sec. 3.1) except that in the Synchronous one, the key-value pair produced at time step $t$ is fed to both KV-memory and FW-memory (i.e., Eq. 12 takes $(\mathbf{k}_t, \mathbf{v}_t)$ as inputs). Figure 1B illustrates the model.

This approach also directly connects to the hybrid model explored in Arora et al. [21], which uses the vanilla LA, whose performance is known to largely lag behind those of more advanced FWPs, such as DeltaNet and GLA [5, 35, 28]. As we demonstrate experimentally, taking into account the latest advances in FWPs/DeltaNet is crucial to study the optimal design choice of HQLT—a critical aspect that has been missing in prior work [21, 22].

## 4 Experiments

We conduct our main experiments using three types of tasks: general language modeling tasks (for general evaluation and sanity checks of HQLTs; Sec. 4.1), synthetic algorithm tasks (to evaluate whether HQLTs can make use of the advantageous computational expressivity of FW-memory to remediate KV-memory's deficiency; Sec. 4.2), and retrieval intensive tasks (to test whether HQLTs can leverage the precise recall ability of KV-memory which is lacking in FW-memory; Sec. 4.3). Further experimental details, including training hyper-parameters and detailed task descriptions, can be found in Appendix A.

### 4.1 Evaluating general language modeling capabilities

We begin with evaluating general language modeling capabilities of the proposed HQLT models as a general sanity check. We train language models with either 340M or 1.3B trainable parameters (using sequence lengths of 2048 and 2240, respectively, unless otherwise noted) from scratch on 15B tokens of the HuggingFace FineWeb-Edu dataset [20]. These choices mostly follow the configurations of the recent work on LTs/DeltaNet [28, 29, 7] and their most recent recommendations available on the `flame` repository [24], including the choice of tokenizer (`fla-hub/transformer-1.3B-100B`). The subword-unit vocabulary size is 32K for all models. All models have 24 layers. The baseline Transformer architecture is from [38] (denoted as Transformer++, following prior convention) and the DeltaNet configuration is from [29]; both models/configurations are used in prior work [29, 7]. Note, however, that for fair comparison with the baseline quadratic transformer, we do not use short-window convolution in the DeltaNet, unlike in recent work [29].

We evaluate the trained models through two perplexity evaluation settings on WikiText-2 [39] (Wiki.) and LAMBADA (LMB.) [40], and six zero-shot common sense reasoning tasks: PiQA [41], HellaSwag (Hella.) [42], WinoGrande [43] (Wino.), ARC-easy (ARC-e) and ARC-challenge (Arc-c) [44]. The choice of these tasks also follow the common settings in prior work [28, 29, 7]. We use the standard `lm-evaluation-harness` [19] for these evaluation runs.

Table 2: General language modeling experiments for Hybrid Quadratic-Linear Transformers (HQLTs). Both the 340M/1.3B models are trained for 15B tokens. Unless otherwise indicated in ablation studies, the window size for KV-memory is 64 tokens, and the dynamic vector mixing is used to combine KV-memory and FW-memory in HQLTs. `lm-evaluation-harness` is used to produce the results.

| Model | Wiki. ppl ↓ | LMB. ppl ↓ | LMB. acc ↑ | PIQA acc ↑ | Hella. acc_n ↑ | Wino. acc ↑ | ARC-e acc ↑ | ARC-c acc_n ↑ | Avg. |
|---|---|---|---|---|---|---|---|---|---|
| *340M params* | | | | | | | | | |
| Transformer++ | 26.5 | 34.9 | **33.9** | 67.6 | 41.0 | 53.7 | 60.2 | 29.0 | 47.6 |
| DeltaNet | 27.6 | 35.0 | 32.8 | 67.1 | 40.8 | 52.6 | 58.5 | 28.8 | 46.8 |
| HQLT | | | | | | | | | |
|   Delayed-Stream | 26.4 | 33.1 | 33.6 | **67.9** | 42.1 | 51.8 | 59.4 | 29.5 | 47.4 |
|   Delayed-Chunk | 26.7 | 29.9 | 33.5 | 66.8 | 42.3 | 50.9 | 61.1 | **30.6** | 47.5 |
|   Synchronous | **26.3** | **29.4** | 33.3 | 66.2 | **42.7** | 53.8 | 61.5 | 29.4 | **47.8** |
| *1.3B params* | | | | | | | | | |
| Transformer++ | 19.8 | 17.9 | 42.6 | 71.0 | 50.3 | 55.8 | 65.2 | 33.2 | 53.0 |
| DeltaNet | 20.6 | 19.9 | 39.3 | 70.1 | 49.5 | 52.5 | 68.5 | 34.2 | 52.3 |
| HQLT | | | | | | | | | |
|   Delayed-Stream | 20.0 | 16.5 | **43.5** | 70.7 | **51.6** | 56.0 | **69.3** | **36.0** | **54.5** |
|   Delayed-Chunk | 20.2 | 16.3 | 41.3 | 71.8 | 50.9 | 55.0 | 67.9 | 35.2 | 53.7 |
|   Synchronous | **19.8** | **15.9** | 42.8 | **72.0** | 51.5 | **56.1** | 68.1 | 33.1 | 53.9 |
| **Ablations with** *340M params* | | | | | | | | | |
| HQLT Synchronous | | | | | | | | | |
|   *sum mixing* | 27.7 | 34.8 | 32.5 | 67.0 | 41.2 | 52.4 | 60.9 | 28.9 | 47.2 |
|   *dynamic vector mixing (25M)* | 26.3 | 29.4 | 33.3 | 66.2 | 42.7 | 53.8 | 61.5 | 29.4 | 47.8 |
|   *w. Linear Attn. (no DeltaNet)* | 33.3 | 114.2 | 21.0 | 63.2 | 36.4 | 51.7 | 53.8 | 26.8 | 42.2 |
|   *window ×2 = 128* | 26.6 | 27.8 | 35.6 | 68.1 | 41.7 | 51.2 | 61.4 | 30.1 | 48.0 |
|   *window ×4 = 256* | 26.7 | 27.7 | 35.4 | 67.0 | 42.2 | 52.7 | 59.8 | 29.0 | 47.7 |
|   *window ×8 = 512* | 27.0 | 28.0 | 35.9 | 66.3 | 41.3 | 53.2 | 60.1 | 29.0 | 47.6 |

**Results.** Table 2 shows the results. We first observe that the two baselines, the quadratic transformer (Transformer++) and DeltaNet perform comparably at both 340M and 1.3B scales on this general evaluation setting; DeltaNet only lags behind slightly. Consequently, we also observe all the hybrid models to also perform similarly on average over the six common sense reasoning tasks, with a slight improvement over DeltaNet (up to 1% absolute on average) which roughly matches the transformer performance. One task where we observe significant improvements by an HQLT is the LAMBADA perplexity evaluation (second column; LMB. ppl): the best HQLT model, the Synchronous variant, achieves about 15% relative improvements over both transformer and DeltaNet baselines, in both 340M and 1.3B-parameter cases. Interestingly in the 1.3B setting, HQLT Delayed-Stream produces the best results on ARC tasks, achieving the best average performace over 6 tasks. Nevertheless, we find all HQLTs to perform favorably compared to the transformer and DeltaNet baselines in the 1.3B case.

**Ablation studies.** Table 2 (bottom part) also provides several ablation studies on HQLT Synchronous. First of all, we confirm that the choice of FW-memory type matters: replacing DeltaNet with the vanilla linear attention [3] (Linear Attn.) yields large performance drop (large average performance drop from 47.8 to 42.2; and a large increase in perplexities, on LMB. in particular). Second, KV-memory's window size (increased from 64, up to 512) has minimal impacts on the general language modeling performance of HQLT. Finally, we also ablate the type of memory mixing strategies: with the exception of the LMB. perplexity evaluation, where dynamic vector mixing outperforms naive sum mixing, the overall performance is very similar on these general evaluation tasks.

## 4.2 Evaluating Expressivity

Here we compare HQLT variants' abilities to leverage expressivity advantages of the FW-memory component, DeltaNet. We evaluate models on two popular regular language recognition tasks, parity and modular arithmetic without brackets, which are well-known tasks that quadratic transformers and other linear recurrent models fail (due to lack of expressivity), while more expressive, DeltaNet performs well [7]. We mainly follow Grazzi et al. [7]'s experimental settings: we use 2-layer and 3-layer models for parity and modular arithmetic, respectively; training and test sequences have

lengths from 3 up to 40, and from 40 to 256, respectively. The window size for all HQLTs is set to 8 (except for Delayed-Chunk which uses 16). We refer to Appendix A.2 for further experimental details.

Table 3 shows the results. As expected, the two "delayed" variants, Delayed-Stream and Delayed-Chunk, fail on these tasks just like the quadratic transformer. In these "delayed" variations, FW-memory has a delay of a few time steps to access an input, corresponding to the quadratic attention window size; this could be too late to leverage FW-memory's capabilities for these tasks requiring state-tracking in "real time". In contrast, Synchronous HQLT has no such limitation and successfully solves these tasks.

As an extra ablation, we confirm that the choice of FW-memory type is important: as expected, even the Synchronous version fails at these tasks, if vanilla LA (Linear Attn.), which has no known expressivity advantage over softmax attention, is used as the FW-memory module instead of DeltaNet.

Table 3: Evaluating Expressivity of Hybrid Quadratic-Linear Transformers (HQLTs) using regular language recognition tasks: Parity and Modular Arithmetic without brackets (Mod Arith). The top-block results are taken from [7]. We show normalized accuracies [%] (0% is chance level).

| Model | Parity acc ↑ | Mod Arith acc ↑ |
|---|---|---|
| Transformer [7] | 2.2 | 3.1 |
| Mamba [7] | 100.0 | 24.1 |
| DeltaNet [7] | 100.0 | 97.1 |
| HQLT | | |
| Delayed-Stream | 3.3 | 27.8 |
| Delayed-Chunk | 2.8 | 1.4 |
| Synchronous | **100.0** | **97.0** |
| *w. Linear Attn.* | 2.5 | 44.5 |

Given that the delayed versions have no other significant advantage over the synchronous version (apart from the conceptual elegance in introducing explicit division of labors over time), we consider this result as a strong argument to prefer the Synchronous variant over other blending strategies for HQLT. This also allows us to conclude that the design of "Infini-attention" from prior work [22], which makes use of the Delayed-Chunk strategy, is sub-optimal in light of expressivity.

## 4.3 Evaluating in-context retrieval abilities

Here we evaluate HQLT's performance on recall-intensive tasks. Following prior work [29, 45], we focus on three tasks: FDA [45], SWDE [46], and SQuAD [47]—tasks on which we observe large performance gaps between the two baselines, quadratic transformer and DeltaNet. Results on these datasets are notoriously known to be sensitive to the evaluation protocol; following the recommendations from prior work [6], we use the original evaluation script from [45] for FDA (instead of `lm-evaluation-harness` [19]). Taking into account the conclusions from the experiments above (Sec. 4.1 and 4.2), we focus on studying HQLT Synchronous. We additionally focus on the compute-budget friendly 340M-parameter setting for this study (we provide a single run for the 1.3B-parameter HQLT with a window size of 64 only).

Table 4: Evaluating Hybrid Quadratic-Linear Transformers (HQLTs) on retrieval-intensive tasks. Both the 340M/1.3B models are trained for 15B tokens. The default window size for KV-memory is 64 in HQLTs. Dynamic mixers (scalar and vector) add 0.4M and 25M parameters, respectively.

| | Window size | SWDE acc ↑ | SQuAD acc ↑ | FDA acc ↑ | Avg. |
|---|---|---|---|---|---|
| *340M params* | | | | | |
| Transformer++ | 2048 | **44.9** | 36.9 | **52.3** | **44.7** |
| Transformer++ | 1024 | 30.4 | 25.5 | 31.2 | 29.0 |
| DeltaNet | - | 18.5 | 25.2 | 8.6 | 17.4 |
| HQLT Synchronous | | | | | |
| *sum mixer* | 64 | 13.3 | 26.2 | 12.6 | 17.4 |
| *dynamic scalar* | 64 | 21.1 | 27.7 | 11.4 | 20.1 |
| *dynamic vector* | 64 | 20.0 | 28.4 | 10.9 | 19.8 |
| *window* ×2 | 128 | 16.9 | 30.5 | 17.9 | 21.8 |
| *window* ×4 | 256 | 18.6 | 35.6 | 15.1 | 23.1 |
| *window* ×8 | 512 | 22.9 | 35.7 | 17.3 | 25.3 |
| *window* ×16 | 1024 | 34.0 | **37.1** | 50.1 | 40.4 |
| *1.3B params* | | | | | |
| Transformer++ | 2048 | **53.7** | 41.5 | 64.7 | **53.3** |
| DeltaNet | - | 32.9 | 29.9 | 23.6 | 28.8 |
| HQLT Synchronous | 64 | 31.9 | 31.2 | 30.2 | 31.1 |

Table 4 shows the results. We first observe that the type of memory output mixer plays a significant role here (second block from top). The *sum mixer* tends to underperform other mixing strategies on SWDE, while both the simple *dynamic scalar* mixing and *dynamic vector* mixing strategies perform similarly; our ablation on the window size (below) is conducted with the dynamic vector mixing strategy.

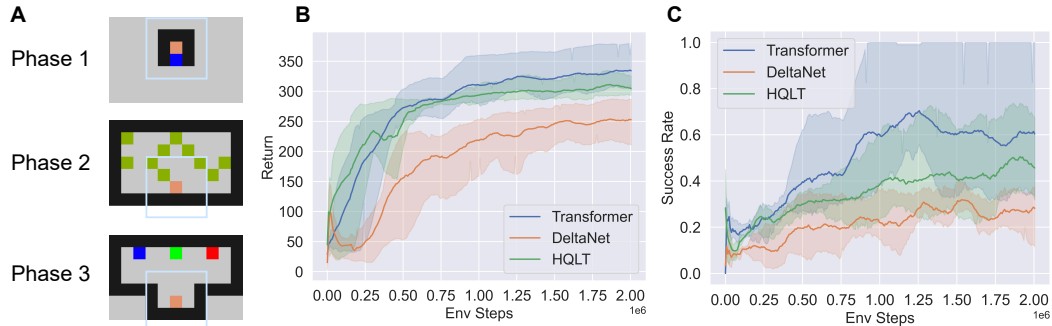

Figure 2: Evaluation of the Synchronous Hybrid Quadratic-Linear Transformer (HQLT) on reinforcement learning in partially observable environments, using a "passive visual match" task [48, 49]. **A:** In this task, an agent (the beige pixel) navigates in a 2D grid world (of size $7 \times 11$) delimited by impermeable walls (black). The agent can only observe the nearby pixels ($5 \times 5$-grid centered on the agent; illustrated by the light-blue boxes). An episode in this task has three phases. During Phase 1 (whose duration is 15 time steps), the agent observes a color, randomly drawn from three choices, red, green or blue (here blue). In Phase 2 (750 steps), the agent is in a room with apples (green); collecting an apple yields a reward of 1. There are initially 10 apples, and they reappear every 20 steps; their positions are random. In Phase 3 (max. 15 steps), if the agent reaches the pixel with the color that matches the one provided in Phase 1, the episode ends successfully; it yields a reward of 100. Alternatively, Phase 3 terminates if the agent reaches a pixel with the wrong color or when the limit of 15 steps is reached (no reward is given in these cases). **B** and **C** show the average return and success rate over 20 test episodes as a function of training environment steps, respectively. We show the average and 95% confidence intervals computed using three training seeds. The variation in success rate is high for Transformer, as one of the seeds consistently achieved the 100% success rate after certain training steps, while other seeds did not. Similarly for HQLT, one of the seeds consistently achieved above 70%.

Next, as expected, the window size is an important parameter for the performance of HQLT on these tasks overall. However, breaking down the performance on the task level, there are mixed results. For example, increasing the window size from 64 to 128 yields large degradation on SWDE (from 20.0 down to 16.9), while increasing it up to 512 yields an effective improvement (SWDE accuracy going up to 22.9). Similarly, increasing the window size from 64 to 128 yields a large improvement on FDA (from 10.9 to 17.9) but increasing it further to 256 yields degradation (down to 15.1). These observations confirm previously reported sensibility of these tasks [6]. Nevertheless, overall, we observe gradual performance increase on *average* by increasing the window size from the default value of 64 to 512, and finally up to 1024. Another interesting comparison is between HQLT with a window size of 1024 and the baseline transformer trained with the same window size (second row); we obtain a large improvement (40.4 vs. 29.0) showing the benefit of an extra FW-memory that covers a larger context, even though retrieval is not the main strength of FW-memory. Finally, by increasing the parameter count to 1.3B, HQLT with a tiny window size of 64 still yields a large improvement over the standalone DeltaNet on FDA. The compute requirements for HQLTs are discussed in Appendix A.1 and B.2.

### 4.4 Evaluation beyond the language domain: Reinforcement learning in POMDPs

To complete the set of experiments above focused on the language domains, here we evaluate HQLT on reinforcement learning (RL) tasks in partially observable environments (POMDPs) [50]. We use the "passive visual match" task [48, 49], whose illustration and description can be found in Figure 2A and its caption. This task is a representative example of memory retrieval with distractions—the task of remembering the initial color after the sub-task of collecting apples in a partially observable room (both tasks require some short-term memory), which is reminiscent of real-life situations, such as remembering where the car is parked after grocery shopping or engaging in a conversation with a friend; these activities act as distractions that can interfere with recalling the parking location.

The sequence length of an episode is 780 steps (15, 750, and 15 steps for each phase, respectively). We compare Transformer (with the full context size of 780), DeltaNet, and Sychronous HQLT (with a short quadratic attention window size of 64). Following Ni et al. [49], we use the hidden size of 100 for all models (we use 2 attention heads). We tested different numbers of layers from 1 to 3

for all models; we found 2 to be the most stable for Transformer, while 3 layers worked best for DeltaNet and HQLT. We train all models using the soft-actor critic method for discrete action space [51, 52] (we have four actions, up/down/left/right, for navigation). Further experimental details can be found in Appendix A.3. Figures 2 **B** and **C** show the results. We observe that the Transformer with softmax attention outperforms DeltaNet on this retrieval-focused task, while HQLT largely closes this performance gap using a short quadratic attention window of only 64.

## 5  Related Work and Discussion

**Related work.** The idea of hybridizing heterogeneous models into a single network is not uncommon [6, 53–61]. A popular approach is the layer-wise combination strategy, which use different model types in different layers in a deep neural network. For example, xLSTM [55] makes use of two types of memory layers, mLSTM and sLSTM; mLSTM is a gated variant of LT/FWP [3, 4], enabling parallel training over sequence elements, while sLSTM enables expressive recurrent computation (as it retains the structure of the classic LSTM [62], with the only distinction being the block-diagonal recurrent structure arising from the multi-head architecture). Other notable examples are Griffin [53] and Samba [57] which interleave gated linear recurrent layers [63–66] and local attention layers.

While such *layer-wise* hybrid methods may be a natural solution to combine memory systems that are structurally very different, e.g., recurrent neural networks and transformer variants, quadratic and linear transformers offer a unique opportunity for blending them at a lower level—within a single layer as an unified memory system, as they share the main variables (namely, key, value, and query vectors) used in their respective computation. Such an opportunity has been previously explored [45, 22] (both of which we related to our models in Sec. 3). In particular, the focus of Arora et al. [45] is on the recall-throughput trade-off—which alone does not cover the full aspects of complementarity between QT and LT, as we've seen; and also crucially, their choice of FW-memory is the vanilla linear attention, whose perform is well-known to lag largely behind the softmax attention (as we also see in our ablation studies replacing DeltaNet with vanilla LA); in contrast, we consider recent advances in LT/FWPs which have developed more competitive LT models, including DeltaNet. Munkhdalai et al. [22] use a form of delta-rule in their FW-memory component; however, recent findings that enhance the delta rule in FWPs [5, 29] seem overlooked (e.g., careful choice of $\phi$, omission of the extra normalizer; use of dynamic learning rate $\beta_t$), which makes it difficult to conclude on the true potential of HQLT. Finally, Munkhdalai et al. [22] use the "Delayed-Chunk" hybrid scheme, which is incompatible with the idea of leveraging FW-memory's expressivity (Sec. 4.2). In this sense, our work offer updated and unique discussions on designing a hybrid attention model which cannot be found in any prior work.

**Limitations.** While this work has shown generally promising results in combining quadratic and linear transformers, the results on the retrieval tasks are still unsatisfactory (Sec. 4.3): it seems that, a large KV-memory window is unavoidable for precise retrieval (even though there was non-zero chance that with enough layers/depth, even a short window could yield a precise long-range retrieval). One potential solution to remediate this may be to introduce a mechanism for more complex communication between KV-memory and FW-memory, in which FW-memory selectively revives certain memory events and reinserts them into the limited KV-memory—as opposed to the current design of sliding window attention in which KV-memory can only operate on the recent past. Investigating such a mechanism is not straightforward in the current era of model development, which sets hardware-efficiency as a hard requirement. Nevertheless, we believe that developing such alternative memory architectures is an exciting direction for future work.

## 6  Conclusion

The quadratic transformer (KV-memory system) and linear transformer (FW-memory system) are two complementary memory systems with unique advantages. We develop a method to blend these two systems into a single hybrid memory system to leverage the strengths of both. By taking into account various types of complementarity between these two systems—complexity, context length, retrieval precision and expressivity, we empirically evaluate our models on general language modeling, retrieval, and synthetic algorithmic tasks and reinforcement learning settings, to finally conclude that the best overall hybrid method is the one that allows KV-memory and FW-memory to operate synchronously, as opposed to alternatives that introduce a division of labor over time. This offers a novel perspective on the development of hybrid memory systems for sequence processing.

## Acknowledgments

The authors are grateful for support from the Kempner Institute for the Study of Natural and Artificial Intelligence, and from the Department of Defense MURI program under ARO grant W911NF-23-1-0277. Kazuki Irie thanks Songlin Yang for a helpful discussion and recommendations regarding the `fla` [24] and `flame` [67] frameworks.

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

# A Experimental details

## A.1 Language modeling experiments

Here we provide details of the language modeling experiments presented in Sec. 4.1 and 4.3.

**Training details.** Table 5 shows the training and model hyper-parameters used to train the 340M and 1.3B parameter models. The KV-memory component of HQLTs use the RoPE positional encodings [68]; Appendix B.3 provides the corresponding ablation study. We train with an effective batch size of 64 per GPU with a sequence length of 2048, using 4 GPUs, for 28,672 steps; this yields 15,032,385,536 tokens. The 1.3B models were trained with a slightly increased sequence length of 2240; this increases the training token count to 16,441,671,680 (for simplicity, in the main text, we refer to both of them as trained for 15B tokens) Note that the actual parameter counts of 340M models are about 370M; we follow this convention from prior work [29, 6, 28, 7] (likely related to the fact that if the models shared the input and output embedding matrices of about 30M parameters, they would have 340M parameters; but this is not the case, either in our work, nor in the prior work).

**Training time.** Training of 340M models using 4 H100-80GB GPUs take about 8 hours for the baseline transformer and 10 hours for DeltaNet and all the HQLT models with the window size from 64 to 1024 tokens. For the 1.3B models, these numbers become 26 hours for the baseline transformer and DeltaNet, and 30 hours for all the HQLTs variants. Given that the training for the 1.3B models uses 16B tokens, training throughputs are 170K tokens/second and 148K tokens/second for the baseline Transformer/DeltaNet and HQLT, respectively.

We use the `fla` [24] toolkit to implement the models, and `flame` [67] to train them.

Table 5: Hyper-parameters of language models.

|  | Model | |
| --- | --- | --- |
|  | 340M | 1.3B |
| Number of layers | 24 | |
| Feedforward block multiplier | 4 | |
| Total hidden size | 1024 | 2048 |
| Number of heads | 8 | 16 |
| Sequence length | 2048 | 2240 |
| Effective Batch size | 64 | |
| Learning rate | $1e^{-3}$ | |
| Warmup steps | 1024 | |
| Minimum learning rate | 0.1 | |
| Max norm clipping | 1.0 | |
| Std. of weight initializers | 0.02 | |

**Implementation details.** Both the Synchronous and Delayed-Stream variants of HQLT studied in this work can directly make use of flash-attention [23] and flash-linear-attention [24] implementations (without modifying the corresponding Triton kernels) for the quadratic and linear components of the models, respectively. For Delayed-Block HQLT, we wrote a custom Triton kernel to replace intra-attention in the DeltaNet implementation of [29] by an efficient softmax attention implementation [23], and modified the backward function accordingly. Note, however, that there is still room for optimization as the intra and inter-chunk operations are currently implemented in two separate kernels (they may potentially be fused into a single kernel for further speed optimization).

**Evaluation Details.** Here we provide further descriptions of the evaluation datasets used in our general language modeling and retrieval tasks.

The six zero-shot common sense reasoning tasks we used in Table 2 are as follows. LAMBADA (LMB.) [40] is a task of predicting the last word in a sentence following some context sentences. PiQA [41] and HellaSwag (Hella.) [42] evaluate models' common sense knowledge (learned in weights) through question answering with multiple choices. WinoGrande [43] (Wino.), inspired by the Winograd Schema Challenge [69], is a set of pronoun-resolution problems. The ARC dataset [44] is a set of grade school-level questions about natural science, which is split into two subsets,

ARC-easy (ARC-e) and ARC-challenge (Arc-c), according to their difficulties (determined based on whether certain baseline models can solve it). We use the standard `lm-evaluation-harness` [19] for evaluation on these datasets.

In Table 4, we use three additional datasets to evaluate models' in-context retrieval abilities. SWDE is an information retrieval task based on the Structured Web Data Extraction dataset [70] (e.g., extracting some subject-predicate-object information from a raw HTML webpage about a movie). Similarly, FDA is an information retrieval task, extracting some key-value pairs from a set of PDFs from the FDA website. SQuAD [47] is the latest version of the Stanford Question Answering Dataset [71] which is a set of reading comprehension problems, which evaluate models' ability to answer question based on a provided text passage. For SQuAD and SWDE, we use `lm-evaluation-harness` [19] for evaluation but by removing leading and trailing spacing in the document (see a related note at `https://github.com/EleutherAI/lm-evaluation-harness/issues/2690`). For FDA, we use the original script from `https://github.com/HazyResearch/prefix-linear-attention/blob/main/lm-eval-harness/prompt_scripts/run_jrt_prompt_hf.sh` (following the recommendation by Yang et al. [6] provided at `https://github.com/NVlabs/GatedDeltaNet?tab=readme-ov-file`).

The choice of all these tasks follow prior work [28, 29, 7].

## A.2 Synthetic algorithmic tasks

Here we provide details about the experiments with two regular languages presented in Sec. 4.2. All the basic settings follow those of Grazzi et al. [7].

**Tasks.** In "Parity", an input is a sequence of zeros and ones, and the task is to determine whether the number of ones in the sequence is odd or even. This essentially corresponds to modulo 2 addition, with the chance-level accuracy of 50%.

In "Modular Arithmetic (Mod Arith)", the task is a modulo 5 addition and multiplication task (without brackets). Each symbol in an input sequence is either a number (from $0$ to $4$, in the module 5 case, which is our setting), or a mathematical symbol (there are five of those: $\{+, -, *, =, \texttt{eos}\}$ where the last symbol is the extra "end-of-sequence" token; which is not necessary for the modulo 5 case but included by convention). Here not only the output, but also the numbers in the sequences, are drawn between $0$ to $4$. This makes the total vocabulary size of 10 with a chance level accuracy of 20%.

**Model configuration.** The model hidden size is set to 128 and the number of heads is 4. For HQLTs, we use the dynamic vector mixing strategy, and the chunk size is set to 8 (except for the Delayed-Chunk variant, we set it to 16 for an implementation reason). We use 2 layers for Parity and 3 layers for Modular Arithmetic. Naturally, the crucial factor 2 is applied after the sigmoid activation on the dynamic learning rate $\beta_t$ in DeltaNet (Eq. 8) to enhance its state-tracking ability [7].

**Training settings.** We train with a batch size of 1024 for 20,000 steps. We search for the best learning rate among $\{5e^{-3}, 1e^{-3}, 5e^{-4}, 1e^{-4}\}$ (the only difference compared to Grazzi et al. [7] is that this list includes $5e^{-3}$ instead of $1e^{-2}$), each with three seeds. We directly measure the validation accuracy to determine the best configuration. In the main result table, Table 3, we report the best/max result among the seeds for the best configuration, while Table 6 shows variability among seeds. This is a standard practice in formal language recognition tasks [12, 10]. We train with sequence lengths from 3 to 40, and validate on sequences of lengths from 40 to 256. Each training run on a single H100 takes about 70 min.

**Evaluation.** We report "normalized accuracies", that is, by denoting the raw accuracy as $A_{raw}$ and the chance level accuracy as $A_{chance}$, we report $(A_{raw} - A_{chance})/(100 - A_{chance})$, where $A_{chance}$ is 50% for parity and 20% for modular arithmetic (modulo 5), such that all the accuracies are scaled to be between 0 and 100, where 0 is chance-level and 100 is perfect accuracy.

## A.3 Reinforcement learning experiments

Here we provide experimental details of the RL experiments presented in Sec. 4.4. Our main settings follow Ni et al. [49]'s (our implementation is based on their code base), which implement the soft-

Table 6: **Median accuracy** and median absolute deviation over 3 seeds using the best learning rate for each case. This is to complete Table 3 which shows the best/max result among the seeds. The top-block results are taken from [7]. We show normalized accuracies [%] (0% is chance level).

| Model | Parity acc ↑ | Mod Arith acc ↑ |
|---|---|---|
| Transformer [7] | $0.3 \pm 1.3$ | $1.8 \pm 0.9$ |
| Mamba [7] | $100.0 \pm 0.0$ | $21.4 \pm 2.7$ |
| DeltaNet [7] | $99.9 \pm 0.6$ | $82.6 \pm 14.6$ |
| HQLT | | |
| Delayed-Stream | $3.0 \pm 0.3$ | $22.2 \pm 5.6$ |
| Delayed-Chunk | $2.3 \pm 0.1$ | $1.4 \pm 0.1$ |
| Synchronous | $99.7 \pm 0.1$ | $93.2 \pm 3.2$ |
| *w. Linear Attn.* | $2.1 \pm 0.3$ | $32.9 \pm 11.6$ |

actor critic method for discrete action space [51, 52]. Regarding the environment, we set a reward of 100 for a successful episode, i.e., when the agent successfully reaches the correct color in Phase 3, instead of 10 in Ni et al. [49].

An observation (a $3 \times 5 \times 5$ image) is first processed by a 2-layer convolutional neural network to produce a 144-dimensional vector, which is then fed to the sequence model (described in the main text). The output of the sequence model is a 100-dimension vector, which is fed to all the policy and value networks; each of which is parameterized by a 2-layer feedforward network with a hidden dimension of 256, and their final output dimension is 4, corresponding to the number of actions.

We use a batch size of 64 and a learning rate of $3e^{-4}$ for all the system components, using the Adam optimizer [72]. We apply a scale of $0.1$ on the entropy term in the loss. Importantly, we use dropout with a dropping rate of $0.1$ inside the sequence models, including on the observation embeddings. We keep the dropout active during rollouts to enhance exploration as is done in Ni et al. [49] (we found this to be important in practice). We refer to the code for any further details.

## B Further discussion

### B.1 Further clarifications for the Delayed-Chunk variant

Delayed-Streaming HQLT and Delayed-Chunk HQLT are not equivalent (i.e., their outputs are different for the same input) because the classic equivalence between recurrent vs. chunk mode computation in linear attention does not hold when softmax is applied within the intra-chunk attention. To be more specific, the Delayed-Streaming variant has a sliding window attention with a stride of 1, i.e., at every time step, the oldest token from the KV-memory window is fed to the FW-memory, and a new token enters the KV-memory window; as a consequence, the memory content of FW-memory is updated at every time step, whereas in the Delayed-Chunk variant, the sequence is processed chunk-by-chunk, i.e., the content of FW-memory is only updated at the chunk-boundaries and remains constant while the system is processing the current chunk. Within a chunk, the content of KV-memory is updated at every time step by adding the new element in the chunk; its content is reset (to empty) at the chunk boundaries. Figure 3 provides an illustration. Delayed-Chunk HQLT is naturally compatible with efficient training: we can apply flash-linear-attention for inter-chunk attention computation, and flash-attention to the intra-chunk attention with softmax.

Intuitively, Delayed-Streaming HQLT may be argued to be a better approach as the KV-memory always makes use of the full window size (except at the very beginning of the sequence), but we also included Delayed-Chunk which is not only a natural extension of chunk-wise parallel form of linear attention, but was used in prior work by Munkhdalai et al. [22].

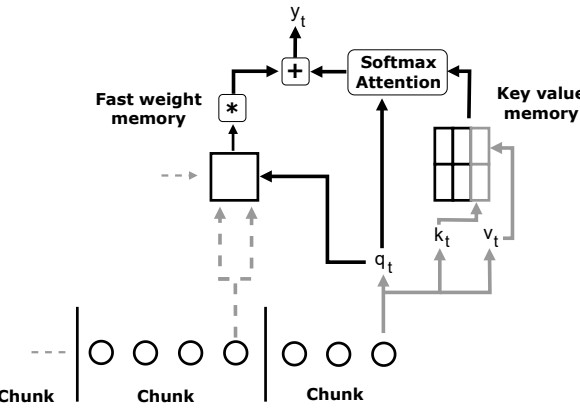

Figure 3: An illustration for the Chunk-Delayed variant of Hybrid Quadratic-Linear Transformers (HQLTs). Here the chunk size is 4. KV-memory only has access to the tokens within the current chunk (i.e., the last three tokens here), whereas FW-memory does not contain any information about the current chunk.

## B.2 Practical computational costs of increasing the window size

In Sec. 4.3/Table 4, we demonstrated how increasing the quadratic attention window size in HQLTs improves their retrieval abilities. Here we discuss the computational costs for such improvements. In practice, within the range of window sizes discussed here, the extra computational cost is negligible, since during inference with a batch size of 1, short-window attention is typically memory-bound rather than compute-bound and can be efficiently computed in a few parallelizable matrix multiplication steps on a GPU, regardless of whether the window size is 64 or 1024. Training time is also only minimally affected by the increased window size as training is parallelized over the sequence elements. However, the total state size (which has a direct impact on the memory requirement) increases as a function of the window size $S$ as $S(d_{\text{in}} + d_{\text{out}}) + d_{\text{out}} * d_{\text{in}}$ per layer and per head.

## B.3 Positional Encodings in KV-memory

All the HQLT models presented in the main text use the RoPE positional encodings [68] in KV-memory. However, it is also known that multi-layer self-attention can process sequences without explicit positional encodings [73–75], and such a "No Pos" approach has certain generalization benefits [12, 76]. Here we provide an ablation study on the positional encoding in KV-memory.

Table 7: General language modeling experiments for Synchronous HQLT. Both the 340M/1.3B models are trained for 15B tokens. "No Pos" indicates no positional encoding in KV-memory. "window" specifies the window size for KV-memory, and the dynamic vector mixing is used to combine KV-memory and FW-memory.

| Model | Wiki. ppl↓ | LMB. ppl↓ | LMB. acc↑ | PIQA acc↑ | Hella. acc_n↑ | Wino. acc↑ | ARC-e acc↑ | ARC-c acc_n↑ | Avg. |
|---|---|---|---|---|---|---|---|---|---|
| *340M params* | | | | | | | | | |
| window = 64 | | | | | | | | | |
| RoPE | **26.3** | **29.4** | **33.3** | 66.2 | **42.7** | **53.8** | **61.5** | 29.4 | **47.8** |
| No Pos | 27.3 | 31.8 | 32.4 | **66.2** | 40.7 | 51.9 | 60.3 | **29.8** | 46.9 |
| window = 1024 | | | | | | | | | |
| RoPE | **26.8** | **31.6** | **34.6** | 66.7 | **41.1** | 50.0 | **59.9** | 28.1 | **46.7** |
| No Pos | 27.2 | 34.5 | 32.3 | 66.3 | 40.2 | **54.0** | 57.3 | **29.4** | 46.6 |
| *1.3B params* | | | | | | | | | |
| window = 64 | | | | | | | | | |
| RoPE | **19.8** | **15.9** | **42.8** | 72.0 | **51.5** | **56.1** | **68.1** | 33.1 | **53.9** |
| No Pos | 20.7 | 17.7 | 40.4 | 70.7 | 50.6 | 54.5 | 67.1 | **33.9** | 52.9 |

Table 8: Evaluating Synchronous HQLT on retrieval-intensive tasks. Both the 340M/1.3B models are trained for 15B tokens. "No Pos" indicates no positional encoding in KV-memory. "window" specifies the window size for KV-memory, and the dynamic vector mixing is used to combine KV-memory and FW-memory.

| | SWDE acc ↑ | SQuAD acc ↑ | FDA acc ↑ | Avg. |
|---|---|---|---|---|
| *340M params* | | | | |
| window = 64 | | | | |
| RoPE | **20.0** | **28.4** | 10.9 | **19.8** |
| No Pos | 17.1 | **28.4** | **13.4** | 19.6 |
| window = 1024 | | | | |
| RoPE | 34.0 | **37.1** | 50.1 | 40.4 |
| No Pos | **36.5** | 33.3 | **55.8** | **41.9** |
| *1.3B params* | | | | |
| window = 64 | | | | |
| RoPE | 31.9 | 31.2 | **30.2** | **31.1** |
| No Pos | **32.7** | **32.9** | 26.3 | 30.6 |

We compare the Synchronous HQLT models with and without RoPE for the 340M models with a window size of 64 or 1024, as well as for the 1.3B model with a window size of 64.

Tables 7 and 8 show the results for general language modeling and retrieval tasks, respectively. Regarding the general language modeling tasks, we find RoPE to be generally beneficial. We observe that RoPE consistently improves the perplexity on Wiki. and LMB. (the two first columns in Table 7); and also on the zero-shot tasks in Table 7, with the exception of Wino. and ARC-c in the 1024 window size case, RoPE yields notable improvements. Another interesting observation here is that the model with a window size of 1024 is slightly worse than the 64-size model in these general language tasks; this may be acceptable given the substantial improvements observed in the retrieval-intensive tasks.

The results are more mixed in the retrieval-intensive tasks (Table 8), which is also likely related to the sensibility of these tasks (as mentioned above). For example, we observe a large improvement on FDA by removing the positional encoding in the 340-parameter models (in both the 64- and 1024-window size cases), while this is not the case in the 1.3B model. In fact, in the 340-parameter 1024-window model, "No Pos" outperforms RoPE on both SWDE and FDA, while the trend is reversed on SQuAD. Nevertheless, given the general language modeling results above, using RoPE seems to be a more robust option overall. Further investigation into the length generalization benefits of "No Pos" HQLTs is beyond the scope of this work.

