# OpenReview forum: "Blending Complementary Memory Systems in Hybrid Quadratic-Linear Transformers"
_NeurIPS.cc/2025/Conference — NeurIPS 2025 poster_

### Official Review · Reviewer_GAJr · 2025-06-22

**Clarity:** 3
**Significance:** 2
**Originality:** 2
**Rating:** 4
**Confidence:** 4

**Summary:**

This paper proposes Hybrid Quadratic-Linear Transformers (HQLTs) that combine key-value (KV) memory from standard transformers with fast-weight (FW) memory from linear transformers, specifically using DeltaNet. The authors argue these systems have complementary properties: KV-memory provides precise retrieval but suffers from quadratic complexity, while FW-memory supports unbounded sequences with higher expressivity but lower retrieval precision. The authors proposed 3 strategies to blend these two systems together: Delayed-Streaming (old KV pairs feed into FW-memory), Delayed-Chunk (chunk-wise processing), and Synchronous (both memories process inputs simultaneously). The authors evaluate on language modelling (340M and 1.3B parameters trained on 15B tokens), synthetic algorithmic tasks (parity, modular arithmetic), and retrieval tasks (FDA, SWDE, SQuAD), concluding that the Synchronous variant performs best.

**Questions:**

1. **On theoretical grounding**: Why invoke "Complementary Learning Systems" when your dichotomy (efficiency vs. precision) fundamentally differs from the established neuroscientific concept (episodic vs. semantic memory with different learning rates)?
2. **On efficiency claims**: Table 4 shows HQLT needs 1024-token windows for competitive retrieval. The appendix claims costs are "negligible" but provides no analysis. Can you provide concrete FLOPs/throughput comparisons? Specifically: HQLT-1024 vs Transformer-2048?
3. **On LAMBADA**: Why does HQLT dramatically outperform on LAMBADA with just 64 tokens when other retrieval tasks fail? This contradicts your other results and deserves investigation.
4. **On synergy**: How do the algorithmic task results demonstrate synergy rather than just "DeltaNet works when included"? What can the hybrid do that neither component could achieve alone?
5. **On comparisons**: Why no comparison to other long-context solutions (Mamba, Transformer-XL, EM-LLM, etc.)? How does HQLT compare to simply using Flash Attention with longer contexts?

_Clear criteria for score improvement_:

- Provide thorough computational analysis showing efficiency gains
- Demonstrate retrieval working with small windows (<256)
- Show true synergistic capabilities beyond individual components
- Correct theoretical framing, better justify your choice or remove CLS references
- Compare against modern long-context baselines

**Ethical Concerns:**

["NO or VERY MINOR ethics concerns only"]

**Final Justification:**

After reviewing the authors' rebuttal and additional experiments, I am raising my score from 3 to 4 and the quality rating. The new HQLT-1024 vs Transformer-1024 results (40.4 vs 29.0) demonstrate genuine complementarity between the memory systems, and the systematic finding that synchronous blending preserves DeltaNet's expressivity while delayed variants don't provides valuable architectural insights for future hybrid models. While practical improvements remain modest and computational trade-offs exist (~20% throughput reduction), the work makes meaningful contributions to understanding how to effectively combine complementary memory systems, which is particularly relevant given the community's current interest in hybrid architectures.

**Limitations:**

The authors acknowledge some limitations (e.g., large windows still needed for retrieval) but don't adequately address:
- Computational overhead of dual memory systems
- Questionable practical advantages over simpler alternatives
- Sensitivity of results to hyperparameters
- Missing comparisons to other long-context solutions

**Paper Formatting Concerns:**

No formatting issues identified.

**Quality:**

3

**Strengths And Weaknesses:**

### Strengths
**Quality:**
- The paper includes a comprehensive experimental evaluation across diverse tasks (language modelling, algorithmic reasoning, retrieval).
- The authors tried different model scales (340M, 1.3B) with consistent training protocols. Of course, more scales could be useful, but I do recognise that this requires very large computation capabilities that are not always available.
- The paper also contains thoughtful ablations on mixing strategies, window sizes, and FW-memory choices.
- The authors provided their source code, which aids reproducibility and is commendable.
- Finally, there is a clear mathematical formulation of the three proposed hybrid variants.

**Clarity:**
- The paper is well-structured with clear motivation and background.
- It contains helpful visual illustrations (Figure 1).
- It also contains explicit equations for each variant

**Originality:**
- The paper updates prior hybrid attempts by using the (relatively) state-of-the-art and very interesting DeltaNet model instead of vanilla linear attention.
- It also includes the systematic exploration of three conceptually distinct blending strategies mentioned above.
- Finally, the evaluation on expressivity tasks to test memory synergy is (to my knowledge) novel.

### Weaknesses
**Quality:** I believe there are a few very important issues here which the authors need to clearly address to have their work accepted in a conference like NeurIPS.
1. **Misappropriation of established terminology**: The first big issue is that the paper invokes "Complementary Learning Systems" (CLS) from neuroscience but proposes an entirely different dichotomy, as far as I understand. Classical CLS (McClelland et al., 1995) describes hippocampal-neocortical differences in learning rates and memory types (episodic vs. semantic), not computational trade-offs between precision and efficiency. The authors do not justify this properly which leads me to believe that it is a misuse of the term, thus raising concerns about theoretical grounding. However, there might be a rationale behind this that is not clear from the current version of the manuscript, in which case I would kindly ask the authors to elaborate.
2. **Core claims not supported by results:**
	* On standard language tasks, improvements are typically at least <1% over baselines. The only notable gain is ~15% on LAMBADA perplexity, which appears task-specific. Given the added architectural complexity, these minimal gains don't justify the approach.
	* Table 4 shows HQLT requires large windows (1024 tokens) to approach Transformer performance. With small windows (64), performance is dramatically worse (19.8 vs 44.7 avg accuracy). This suggests the hybrid isn't truly leveraging complementary strengths but rather defaulting to brute-force quadratic attention.
3. **Missing critical computational analysis**: Despite computational efficiency being a primary motivation, the paper provides no FLOPs analysis or throughput comparisons. The appendix dismisses this with "the actual computational cost is negligible" (line 459), which is inadequate. How does HQLT with 1024-token window compare to Transformer++ with 2048 tokens?
4. **The expressivity results don't demonstrate synergy**: Synchronous HQLT solving parity/modular arithmetic merely shows DeltaNet works when included (see Table 3 and Appendix Table 7), not that the hybrid achieves something neither component could alone.

**Clarity:** (Minor issues)
- Inconsistent terminology (HQLT vs HQTL)
- The LAMBADA anomaly (strong performance with small windows) is unexplained
- Defers crucial efficiency discussion to appendix

**Significance:**
1. **Limited practical impact**: If large windows are still required for retrieval, the hybrid approach fails to solve the fundamental limitation it targets
2. **Algorithmic task results are unsurprising**: As mentioned above, showing that Synchronous HQLT (which includes DeltaNet) solves tasks that DeltaNet already solves doesn't demonstrate synergy.
3. **No comparison to other long-context solutions** (Transformer-XL, EM-LLM, Compressive Transformer, etc.)

**Originality:**
- Core hybridization concept exists in prior work (Arora et al., Munkhdalai et al.)
- Main distinction is using DeltaNet, which feels incremental
- The successful "Synchronous" approach is conceptually straightforward

---

> ### Author Rebuttal · Authors · 2025-07-30
>
> We sincerely thank the reviewer for their valuable and thorough feedback. We believe there are a few factual misunderstandings that have resulted in the given score, which we’d like to resolve. Overall, we believe that we have good explanations to resolve all the concerns.
>
> > 1. (Misappropriation of established terminology & Question 1) Why invoke "Complementary Learning Systems" ?
>
> The decision of referencing CLS was carefully discussed among the authors; we justify it as follows. First of all, we very explicitly state that the complementarity we discuss is **different** from those of the classic CLS; please see Line 33: “Here we propose a **different division of labor** into complementary linear and quadratic memory systems each with complementary strengths (Table 1).”
>
> We still concluded that “complementary” is an appropriate/compelling term to use when contrasting the two transformer variants.  The fundamental question we ask here is (Line 26): “How could we achieve multiple computational properties that are incompatible within a single system?”  Under this context, we invoke CLS, because, beyond the specific nature of complementarity represented in CLS, it also represents an important example of a high-level principle of achieving multiple incompatible goals by combining complementary systems. We find it inappropriate to dismiss such an important prior discussion on complementarity.
>
> If the reviewer still believes that there are misleading sentences that may lead readers to believe we conflate the specific complementarity we discuss with that of CLS, we kindly ask the reviewer specify which sentences are problematic.
>
> > 2. (Core claims not supported by results:) On standard language tasks, improvements are typically at least <1% over baselines. The only notable gain is ~15% on LAMBADA perplexity, which appears task-specific. Given the added architectural complexity, these minimal gains don't justify the approach.
>
> There seems to be some misunderstanding. We have no such “core claims” based on the experiments of Table 2. On the contrary, we very explicitly stated that these general language modeling experiments are sanity checks: see in Line 202: “We conduct our main experiments using three types of tasks: general language modeling tasks (for general evaluation and **sanity checks** of HQLTs; Sec. 4.1)” as well as in Line 210 “We begin with evaluating general language modeling capabilities of the proposed HQTL models as a **general sanity check.**”.
>
> > (2. continued) Table 4 shows HQLT requires large windows (1024 tokens) to approach Transformer performance. With small windows (64), performance is dramatically worse (19.8 vs 44.7 avg accuracy). This suggests the hybrid isn't truly leveraging complementary strengths but rather defaulting to brute-force quadratic attention.
>
> We would like to respond to this in two steps. First, we conducted extra experiments by training one more 340M-param baseline transformer with a context window size of 1024 (as opposed to 2048 in the main baseline). We compared this KV-memory-only baseline to the Synchronous HQLT model with sliding window size 1024 (reported in the submission). We obtained the following results on the retrieval-intense tasks of Table 4:
>
> | Model | KV window size |  SWDE | SQuAD | FDA | Avg |
> |-------------|----------------|-------------|----------------|-------------|----------------|
> | Transformer ++ | **1024** | 30.4 | 25.5 | 31.2 | 29.0 |
> | Synchronous-HQLT | **1024** | 34.0 | 37.1 | 50.1 | **40.4** |
>
> Here we see that the combination of KV- and FW-memory effectively improves over the KV-only system, showing that FW-memory can potentially help KV-memory in retrieval even though it is not its main strength.
>
> Second, we clarify that we made no claim regarding such a “synergy”: we clarified upfront that the strength of KV-memory is retrieval, and FW-memory’s strength is expressivity. In this regard, we clearly commented that (Line 354): “the results on the retrieval tasks are still unsatisfactory (Sec. 4.3): it seems that a large KV-memory window is unavoidable for precise retrieval”). Therefore, we respectfully disagree with the reviewer’s claims that our “Core claims are not supported by results”. All our claims are based on our experimental results, and we honestly discussed limitations where appropriate (Reviewer Z43k highlighted this among our strengths).
>
> > 3. (Missing critical computational analysis & Question 3): Despite computational efficiency being a primary motivation, the paper provides no FLOPs analysis or throughput comparisons. The appendix dismisses this with "the actual computational cost is negligible" (line 459), which is inadequate.
>
> First of all, computational efficiency is not our “primary motivation”. This is an important misunderstanding to be resolved. The core essence of this study is that **we have multiple goals**: one is indeed to minimize long context quadratic attention, another one is to obtain strong expressivity, and yet another one is to enable precise retrieval. Our main claim is that these goals cannot be all achieved by an individual system. Building and studying hybrid systems to simultaneously achieve these normally incompatible goals is our primary motivation.
>
> Secondly, we kindly ask the reviewer to consider the full sentence (not just a partial segment which is misleading). The full sentence was (Line 459 in the appendix): “In practice, within the range of window sizes discussed here, the actual computational cost is negligible, since during inference with a batch size of 1, short-window attention can be efficiently computed with a few/two steps of parallelizable matrix multiplication on a GPU, regardless of whether the window size is 64 or 1024. Training time is also only minimally affected by the increased window size as training is parallelized over the sequence elements.” etc… We did not “dismiss” it.
>
> The run-time comparison was reported in Appendix A.1 (Line 386): “Training of 340M models using 4 H100-80GB GPUs take about 8 hours for the baseline transformer and 10 hours for DeltaNet and all the HQLT models” etc… Given that the 340M models are trained using 15B tokens, all the information was available to compute the throughputs. Regarding Transformer-2048 vs. HQLT-1024, their throughputs are 520K tokens/second vs 417K tokens/second, respectively. We agree it is unfortunate that there is no gain in efficiency overall (we made no such a claim), but in exchange, the resulting hybrid can perform expressive computation that transformers cannot (Table 3), and do better in retrieval than DeltaNet (Table 4).
>
> >4. (The expressivity results don't demonstrate synergy & Question 4): Synchronous HQLT solving parity/modular arithmetic merely shows DeltaNet works when included (see Table 3 and Appendix Table 7), not that the hybrid achieves something neither component could alone.
>
> That is our conclusion. We never claimed that we expect “synergy” between the two systems in terms of expressivity. We are currently not aware of any formal language tasks on which transformers outperform DeltaNet (even though such an investigation is an interesting future research direction). We clarified upfront that expressivity is FW-memory’s strength that is lacking in KV-memory.
>
> > (Question 4, part 2) What can the hybrid do that neither component could achieve alone?
>
> This is an excellent open question. We are currently not aware of tasks on which both FW-memory and KV-memory fail systematically. We believe this is a very important question for future work.
>
> > (Question 3) On LAMBADA: Why does HQLT dramatically outperform on LAMBADA with just 64 tokens when other retrieval tasks fail? This contradicts your other results and deserves investigation.
>
> There is another misunderstanding: LAMBADA is not a retrieval task. None of the tasks in Table 2 is. It is a general text completion task relying much more on the recent contexts compared to the retrieval-intensive tasks studied in Table 4. There is nothing contradictory about this result.
>
> > (Question 5) On comparisons: Why no comparison to other long-context solutions (Mamba, Transformer-XL, EM-LLM, etc.)? How does HQLT compare to simply using Flash Attention with longer contexts?
>
> Our work contains neither claims nor experiments regarding long-context processing, which is well-studied in prior work on linear attention/DeltaNet, including comparison to Mamba and Transformer-XL; see e.g., [5, 23, 15]. All our language experiments are evaluated with the length of 2048 (and as we specified, all the quadratic attention implementations use FlashAttention2 [16]).
>
> > (Clear criteria for score improvement): Demonstrate retrieval working with small windows (<256)
>
> We already reported that the short window size of 256 is not enough for the retrieval tasks studied in Table 4. Here we conducted an extra experiment to evaluate the HQLT on reinforcement learning in partially observable environments involving retrieval (due to the rebuttal length limitation, we kindly ask the reviewer to check the ADDITIONAL NOTE in our response to Reviewer **Mb9d** for further details). There, we demonstrate strong retrieval performance of HQLT using a small window size of **64** compared to a transformer with the full context access of length 780.
>
> > (Limitations:) Sensitivity to hyperparameters
>
> We’d like to remind the reviewer that unfortunately, this is not specific to our work. We are not aware of any (recent) work on language modeling at this scale which shows hyper-parameter sensitivity w.r.t., e.g., learning rate.
>
> We hope our responses above will, at very least, resolve the core misunderstandings. We provided our thorough response to all the critiques/criteria for score improvement raised by the reviewer.  If the reviewer found our response useful, we would be grateful if the reviewer would consider raising the score. In any scenario, we thank the reviewer again for their thorough and thoughtful review.

---

> > ### Comment · Reviewer_GAJr · 2025-08-03
> >
> > I'd like to thank the authors for their detailed response and additional experimental results. After considering all clarifications and the other reviewers' comments, I have a few points to update in my assessment:
> >
> > * The new HQLT-1024 vs Transformer-1024 comparison (40.4 vs 29.0 avg) provides indeed important evidence that FW-memory contributes to retrieval when controlling for window size. This addresses my concern about whether the hybrid leverages true complementarity.
> > * The throughput data (520K vs 417K tokens/sec) helps quantify the computational trade-offs, though I still believe that a full 'FLOPs' analysis would strengthen the camera-ready version.
> > * Your explanation that Table 2 represents "sanity checks" rather than core claims helps contextualise those results appropriately.
> >
> > In addition, there are points that I think still remain partially addressed:
> >
> > * Regarding the CLS framing, while you are correct that you note the distinction in Line 33, invoking CLS still seems unnecessary given your different dichotomy. However, I do understand it is still an interesting connection and might steer future computational models of cortico-hippocampal interactions to examine this direction too. Given this, please consider either strengthening a bit the connection by elaborating slightly more or using an alternative framing.
> > * The ~20% throughput reduction is non-trivial. While you clarify efficiency wasn't the "primary" goal, the paper's motivation around addressing quadratic complexity suggests users should be clearly informed of these trade-offs.
> > * Outside of specific scenarios (LAMBADA, synthetic tasks), the improvements remain modest, which affects adoption potential.
> >
> >
> > Some points that I value upon reconsideration:
> >
> > * The systematic demonstration that synchronous blending preserves DeltaNet's expressivity while delayed variants don't is a valuable finding for future hybrid model design.
> > * Your systematic evaluation with modern FW-memory (DeltaNet) meaningfully updates prior hybrid attempts that used vanilla linear attention.
> > * Your transparent discussion of limitations (e.g., large windows still needed for certain retrieval tasks).
> >
> > Given these additional evidence and clarifications, I am raising my score from 3 to 4 (borderline accept). I think that given the community's current interest in hybrid memory, the paper provides valuable insights despite the modest empirical gains. In case it is accepted, some minor suggestions for the camera-ready version are:
> >
> > * To add an explicit FLOPs comparison table
> > * To clarify the CLS connection or consider alternative framing
> > * To highlight the synchronous vs. delayed finding more prominently as a key contribution

---

> > > ### Author Response · Authors · 2025-08-04
> > >
> > > We sincerely thank the reviewer for this detailed response, thorough consideration of our response, and the increased score. We will ensure that all the discussions and points raised here are taken into account as we work to improve our final revision. Thank you for all the excellent suggestions that help us improve our paper.

---

### Official Review · Reviewer_Z43k · 2025-06-30

**Clarity:** 3
**Significance:** 3
**Originality:** 2
**Rating:** 4
**Confidence:** 4

**Summary:**

This paper introduces Hybrid Quadratic-Linear Transformers (HQLTs), a new class of sequence models that blend two complementary neural memory systems: quadratic key-value (KV) memory, known for precise retrieval but limited context due to quadratic complexity, and linear fast-weight (FW) memory, known for unbounded context and expressivity but imprecise recall. The authors explore three methods to integrate these systems - Delayed-Streaming, Delayed-Chunk, and Synchronous - and conduct extensive experiments to evaluate them across language modeling, retrieval, and synthetic algorithmic tasks. Results show that the synchronous variant, which allows both memory systems to operate concurrently on the same input, outperforms others by effectively combining the strengths of each memory system. The work establishes that such hybridization not only overcomes individual limitations but also enables new capabilities in memory system design, especially when advanced FW memory models like DeltaNet are used.

**Questions:**

1. Can you show a clear case where hybridization gives a meaningful win over both standalone Transformer and DeltaNet, even with tuning or scaling? This would help justify the added complexity.

2. The choice between sum, scalar, and vector mixing seems arbitrary. What guides this design? Is there a principled way to adapt mixing across tasks or contexts?

3. Retrieval tasks still rely heavily on the KV-memory window size. Could you explore a mechanism where FW-memory actively contributes to retrieval, not just as a backup store?

4. Comparison to other recent hybrids like Infini-attention or Mamba-based models is missing. Can you provide head-to-head results to clarify what your method uniquely improves?

**Ethical Concerns:**

["NO or VERY MINOR ethics concerns only"]

**Final Justification:**

I appreciate the author's clarifications and have adjusted my evaluation upward accordingly.

**Limitations:**

yes

**Paper Formatting Concerns:**

-

**Quality:**

3

**Strengths And Weaknesses:**

Quality: The paper is technically solid and presents a complete piece of work. The proposed hybrid models are carefully evaluated across a wide range of tasks, and the use of advanced fast-weight methods like DeltaNet adds credibility. The authors are honest about limitations and support their claims with solid empirical evidence. However, the performance improvements are sometimes small, and the architectural blending techniques feel more incremental than transformative. Some choices, like gating mechanisms, would benefit from further theoretical justification.

Clarity: The writing is mostly clear, and the paper is well-organized. The motivation and high-level ideas are easy to follow, but certain sections - especially those describing the hybrid mechanisms - are dense and could be made more accessible. The notation can be overwhelming at times, and clearer visualizations or example-driven explanations would help. Overall, the reader can understand the core contributions, but digesting the technical details takes effort.

Significance: The paper tackles a meaningful problem in transformer architecture design - merging long-context capability with precise memory. The proposed models are relevant to practical scenarios and could influence future work on efficient attention. That said, the real-world impact is still limited by modest gains and unresolved issues with retrieval. The contribution is useful, but not groundbreaking, and the adoption by the broader community will depend on future extensions.

Originality: The idea of combining quadratic and linear attention is not new, but this paper brings a more thorough and updated evaluation, especially by using stronger fast-weight models. The synchronous design and detailed empirical analysis are clear advances over prior work. Still, the methodological novelty is limited - the core ideas are largely recombinations of known mechanisms. The paper is more valuable for its careful integration and insight than for introducing radically new techniques.

---

> ### Author Rebuttal · Authors · 2025-07-30
>
> We thank the reviewer for their valuable feedback and overall positive comments. We sincerely appreciate your thoughtful and high-quality review. We believe that we have good explanations to resolve the reviewer’s remaining concerns. Please find our responses below.
>
> > (Quality) … Some choices, like gating mechanisms, would benefit from further theoretical justification.
>
> > (Question 2) The choice between sum, scalar, and vector mixing seems arbitrary. What guides this design? Is there a principled way to adapt mixing across tasks or contexts?
>
> While we have no "theoretical justifications”, the choices of gating/mixing mechanism studied here are not arbitrary: the “sum” and “scalar mixing” strategies are those used by the closest prior works on hybrid models by Arora et al. [14] and Munkhdalai et al. [15], respectively. (However, based on the prior work, it is unclear what’s the impact of these choices). The last variant, “dynamic vector mixing” is another classic way to combine two pieces of information in deep learning in the style of LSTM-gating. We hope this explanation will resolve the reviewer’s main concern regarding the “arbitrariness” of the gating mechanisms chosen to be studied here.
>
> > (Clarity) The writing is mostly clear, and the paper is well-organized. The motivation and high-level ideas are easy to follow, but certain sections - especially those describing the hybrid mechanisms - are dense and could be made more accessible. The notation can be overwhelming at times, and clearer visualizations or example-driven explanations would help.
>
> Thank you very much for sharing this valuable opinion. One challenge we are facing here is that further simplifying the notations would result in ambiguous mathematical model descriptions. We also tried our best to intuitively illustrate the model in Figure 1. Given that there are other reviewers who seem to like our thorough descriptions/equations (e.g., Reviewer GAJr), we kindly ask the reviewer to confirm (and potentially tell us) if there are specific parts of our description or figure that have room for improvement in terms of accessibility/clarify. We will be more than happy to address the corresponding writing following the reviewer’s suggestions.
>
> > (Significance) … That said, the real-world impact is still limited by modest gains and unresolved issues with retrieval. The contribution is useful, but not groundbreaking, and the adoption by the broader community will depend on future extensions.
>
> We fully acknowledge this limitation. As the reviewer also highlighted positively in the evaluation  of Originality as “The paper is more valuable for its careful integration and insight than for introducing radically new techniques.”, we believe our conceptual contribution has significance for future work as we highlighted how a specific choice of hybrid strategy may have significant impact on the hybrid’s abilities to leverage certain properties of individual systems, here by focusing on the expressivity of FW-memory (Table 3). Also, to the best of our knowledge, no prior work discusses the complete picture of complementarity between quadratic/linear attention that includes expressivity, as is done in our Table 1.
>
> Furthermore, to introduce one additional contribution as an empirical work, we conducted an additional experiment to evaluate the Synchronous HQLT on **reinforcement learning (RL) in partially observable environments**, using the “passive visual match” task [Rebuttal Ref. 1].
>
> [Rebuttal 1] Ni et al. When do transformers shine in RL? Decoupling memory from credit assignment. NeurIPS 2023.
>
> This allows us to validate our method on a domain outside of the language, and also generally highlight the relevance of hybrid models outside of the language modeling world. The task is illustrated in Figure 2 of the corresponding paper [Rebuttal Ref. 1] (we are referring to this paper, because this year, the NeurIPS review system does not allow us to provide our own figures and plots during the rebuttal): In this task, an agent navigates in a 2D grid world (of size 7×11) delimited by impermeable walls. The agent can only observe the nearby pixels (5 × 5-grid centered on the agent). An episode in this task has three phases. During Phase 1 (whose duration is 15 time steps), the agent observes a color, randomly drawn from three choices, red, green or blue. In Phase 2 (750 steps), the agent is in a room with apples; collecting an apple yields a reward of 1. There are initially 10 apples, and they reappear every 20 steps; their positions are random.  In Phase 3 (max. 15 steps), the agent is placed in a room with three colors; if the agent reaches the pixel with the color that matches the one provided in Phase 1, the episode ends successfully; it yields a reward of 100. Alternatively, Phase 3 terminates if the agent reaches a pixel with the wrong color or when the limit of 15 steps is reached (no reward is given in these cases).
>
> The episode/sequence length for this task is maximum 780. The attention window size for the baseline quadratic transformer is set to 780 such that it covers the full episode sequence. We set the KV-memory window size of HQLT to 64 (i.e., less than 10 times the transformer window size). We obtain the following result in terms of average return (over 3 seeds):
>
> |  Model                |  Average Return |
> |-------------|----------------|
> | DeltaNet    |   253.7 |
> | HQLT         |   311.4 |
> | Transformer  |  335.1 |
>
> In the final version, we will include the corresponding learning curves with 95% confidence interval, as is typically done to report RL experiments. We believe that including this RL experiment will broaden the scope of the work and make it more appealing to the wider NeurIPS community.
>
> > (Question 1) Can you show a clear case where hybridization gives a meaningful win over both standalone Transformer and DeltaNet, even with tuning or scaling? This would help justify the added complexity.
>
> Thank you for this question. First of all, we’d like to justify the added complexity by achieving a single model that can simultaneously perform expressive computation that Transformers cannot, and do better retrieval than DeltaNet. Nevertheless, our response to Question 3 below provides evidence for the case where HQTL outperforms both individual Transformer and DeltaNet.
>
> > (Question 3) Retrieval tasks still rely heavily on the KV-memory window size. Could you explore a mechanism where FW-memory actively contributes to retrieval, not just as a backup store?
>
> This is an excellent question. We conducted extra experiments to examine if FW-memory could also potentially complement KV-memory in retrieval tasks. We trained one more 340M-param baseline transformer with a context window size 1024 (as opposed to 2048 in the main baseline). We compared this KV-memory-only baseline to the Synchronous HQLT model with sliding window size 1024. We obtained the following results on the retrieval-intensive tasks of Table 4:
>
> | Model | KV window size |  SWDE | SQuAD | FDA | Avg |
> |-------------|----------------|-------------|----------------|-------------|----------------|
> | Transformer ++ | **1024** | 30.4 | 25.5 | 31.2 | 29.0 |
> | Synchronous-HQLT | **1024** | 34.0 | 37.1 | 50.1 | **40.4** |
>
> Here we observe that HQLT with FW-memory outperforms Transformer with only KV-memory, when both models have the same KV-memory window size. While this remains a small-scale experiment, we believe this is a highly encouraging empirical result demonstrating that FW-memory can potentially behave more than just as a “backup storage” for retrieval, when its span covers longer context than the standalone KV-memory. We will add this discussion in the final revision. We really appreciate the reviewers’ comment prompting us to conduct these useful experiments.
>
> > (Question 4) Comparison to other recent hybrids like Infini-attention or Mamba-based models is missing. Can you provide head-to-head results to clarify what your method uniquely improves?
>
> We thoroughly discussed this in our discussion section (from Line 337 and forth; note that Munkhdalai et al. [15] is the Infini-attention paper). Infini-attention can be obtained in our modeling framework by replacing DeltaNet with either a vanilla linear transformer or a sub-optimal version of DeltaNet (sub-optimal in light of recent advances in DeltaNet [5, 23] as we explained in detail in Line 337) and by using the Delayed-Chunk strategy. As we have shown, such a setting yields a strictly weaker model than our best Synchronous model (see Table 3). Similarly, Mamba is strictly more limited than DeltaNet in terms of expressivity (we did report Mamba performance from Grazzi et al [8] in Table 3), and Mamba also has been shown to underperform other advanced models such as (Gated) DeltaNet in prior work (see [6] and [23]) etc. as a standalone sequence model. Therefore, to save compute-resource, we eliminated these weak baselines that are known to be conceptually and/or empirically less powerful than the models discussed here based on results from well-known prior papers.
>
> We hope our responses above have successfully resolved the remaining concerns of the reviewer. If so, we would be grateful if the reviewer would consider raising the score. Thank you again for all the excellent suggestions that help us improve the final version of the paper.

---

### Official Review · Reviewer_Ww5m · 2025-07-01

**Clarity:** 3
**Significance:** 4
**Originality:** 3
**Rating:** 5
**Confidence:** 5

**Summary:**

This paper addresses the well-known limitations of quadratic and linear attention mechanisms in sequence modeling. The quadratic attention (i.e., standard softmax-based key-value attention) provides precise token retrieval but suffers from quadratic time and memory complexity with respect to sequence length. In contrast, linear attention (i.e., fast-weight programmer) scales linearly with sequence length and supports long-context processing, but struggles to precisely recall specific tokens, especially when the number of tokens to remember exceeds the hidden state dimension. To tackle these limitations, this paper proposes **Hybrid Quadratic-Linear Transformers (HQLTs)**, which combine the precise retrieval capabilities of quadratic attention and the expressive state-tracking of linear attention (specifically, DeltaNet). In particular, this paper introduces and compares three blending strategies (Delayed-Streaming, Delayed-Chunk, and Synchronous) and memory mixing and gating mechanisms to effectively integrate KV-memory and FW-memory outputs. Comprehensive experiments on language modeling, retrieval, and algorithmic tasks demonstrate the potential of HQLTs to mitigate the limitations of each memory system and offer new insights into the design of hybrid neural memory architectures.

**Questions:**

- **Q1**. Regarding the W1, could the authors clarify what distinguishes HQLT from existing hybrid models that also combine quadratic and linear attention mechanisms?
- **Q2**. This paper blends quadratic and linear attention within a single layer. Could the authors elaborate on why this within-layer hybridization is preferable to a layer-wise separation strategy (e.g., using quadratic attention in earlier and linear attention in later layers, or vice versa)?
- **Q3**. Is there potential for an intermediate approach with partial overlap between the Delayed-Streaming (no overlap) and Synchronous (full overlap) strategies? I would like to hear the authors’ perspective on this idea, including any expected benefits and drawbacks.

**Ethical Concerns:**

["NO or VERY MINOR ethics concerns only"]

**Final Justification:**

The response adequately addresses the main concerns raised in the initial review. Accordingly, I have decided to retain my original score.

**Limitations:**

The authors adequately addressed the limitations of their work.

**Paper Formatting Concerns:**

I don't see any major formatting issues in this paper.

**Quality:**

3

**Strengths And Weaknesses:**

### Strengths
- **S1**. The paper clearly motivates the complementarity of quadratic and linear transformers, drawing an interesting connection to the theory of complementary learning systems.
- **S2**. It systematically explores multiple blending strategies that are possible architectural designs for hybridization and empirically evaluates them on diverse tasks, including language modeling, retrieval, and algorithmic benchmarks.
- **S3**. The proposed HQLT framework is compatible with any linear attention variant, making it extensible to future advances in fast-weight or linear attention methods.
- **S4**. The broad range of experiments demonstrates the effectiveness of the hybrid model compared to using either quadratic attention alone or linear attention alone.

### Weaknesses
- **W1**. The concept of hybrid approaches that combine quadratic and linear attention is not entirely novel; prior works such as [1] have explored similar ideas.
- **W2**. As shown in Table 3, on specific tasks (e.g., parity and modular arithmetic), all HQLT variants underperform compared to standalone DeltaNet, suggesting that hybridization can sometimes weaken the strengths of the fast-weight memory.
- **W3**. While the Synchronous HQLT generally outperforms other variants, it is notable that in the 1.3B setting, the Delayed-Stream variant achieves better results on the ARC tasks. This indicates that we could face additional difficulty finding the most appropriate blending strategy for a given use case.

[1] Transformer quality in linear time, ICML 2022

---

> ### Author Rebuttal · Authors · 2025-07-30
>
> We thank the reviewer for many positive comments and their valuable feedback. We sincerely appreciate your thoughtful and high-quality review. We believe that we have good explanations to resolve the reviewer’s remaining concerns. Please find our responses below.
>
> > W1. The concept of hybrid approaches that combine quadratic and linear attention is not entirely novel; prior works such as [1] have explored similar ideas.
>
> > Q1. Regarding the W1, could the authors clarify what distinguishes HQLT from existing hybrid models that also combine quadratic and linear attention mechanisms?
>
> Yes, we fully acknowledge that several prior works have discussed such hybrid modeling (including the reviewer’s reference [1] which is Ref. [25] in our work). Here, we contribute unique perspectives to hybrid strategies by introducing "expressivity" (see Table 1) as a new dimension for building hybrid models using the latest FW-models (no prior work on hybrid models have such a discussion), and how that can affect the precise choice of the hybridizing strategies (the choice made in prior work is arbitrary; e.g., Arora et al. [14] used the Sychronous approach, while Munkhdalai et al. [15] used Delayed-Chunk; but no evidence or explanation was provided regarding this choice). Similarly, we also discussed various mixing strategies, including “sum” (Arora et al. [14]) vs. “scalar mixing” (Munkhdalai et al. [15]). Overall, we provide more comprehensive and updated discussions on building hybrid models (Reviewer Z43k also highlighted this). We believe this is important given the increasing interest in developing hybrid models.
>
> > W2. As shown in Table 3, on specific tasks (e.g., parity and modular arithmetic), all HQLT variants underperform compared to standalone DeltaNet, suggesting that hybridization can sometimes weaken the strengths of the fast-weight memory.
>
> Thank you for pointing this out. There was some sub-optimality with our hyper-parameter search space at the time of submission when reporting results in Table 3. We detected and fixed this problem while working on the supplemental materials. As we reported there (Table 7 in Appendix), after fixing this sub-optimal hyper-parameter search, the performance of HQLT matched the DeltaNet performance reported by prior work by Grazzi et al. [8] on these tasks: Synchronous HQLT achieves 100.0% and 97.0% accuracies on the parity and modular arithmetic tasks, respectively. We believe this update directly resolves the reviewer’s concern.
>
>
> > W3. While the Synchronous HQLT generally outperforms other variants, it is notable that in the 1.3B setting, the Delayed-Stream variant achieves better results on the ARC tasks. This indicates that we could face additional difficulty finding the most appropriate blending strategy for a given use case.
>
> This is a valid point and we agree with the reviewer. However, the Delayed variants are fundamentally limited in their ability to leverage the expressivity advantages of FW memory (Table 3). More generally, we expect that performance differences between these model variants on the general language modeling results may vary depending on the specific choice of KV-memory (e.g., regular softmax attention vs. forgetting attention [Rebuttal Ref 2]) and FW-memory (e.g., Gated DeltaNet [6]) that will be used in the future. In contrast, we expect the importance of being Synchronous to leverage the expressivity strength of FW-memory will broadly remain a valid conclusion for any model types. Therefore, our work demonstrates a strong reason to prefer the Synchronous variant overall (which can be an important result for future work on hybrid models).
>
> [Rebuttal Ref 2] Lin et al. Forgetting Transformer: Softmax Attention with a Forget Gate. ICLR 2025
>
> > Q2. This paper blends quadratic and linear attention within a single layer. Could the authors elaborate on why this within-layer hybridization is preferable to a layer-wise separation strategy (e.g., using quadratic attention in earlier and linear attention in later layers, or vice versa)?
>
> There are two conceptual motivations for within-layer blending. One is that the alternative, layer-wise separation strategy requires human ingenuity to decide where to put which type of memory models, whereas here in the within-layer strategy, the model can flexibly learn to use both types of memory systems in any layer in a context dependent fashion. We believe this yields a more general-purpose layer.
>
> Additionally, for the quadratic-linear transformer hybrids, the two types of transformers are both based on query/key/value variables which can be naturally shared by the two memory systems within the same layer; this is conceptually elegant, and allows us to create hybrid models with a parameter count similar to the non-hybrid models. These explanations can also be found in our Discussion section 5 (paragraph Line 327).
>
> > Q3. Is there potential for an intermediate approach with partial overlap between the Delayed-Streaming (no overlap) and Synchronous (full overlap) strategies? I would like to hear the authors’ perspective on this idea, including any expected benefits and drawbacks.
>
> This is a very interesting question. One conceptually sound approach (which we briefly mentioned in Sec 5. Line 353 Limitations) would be to introduce a “memory router” mechanism that dynamically decides, at every time step, which key/value memory pair should be sent to FW-memory (with or without delay), and which should remain in the KV-memory short window. We expect such a more sophisticated communication between KV-memory and FW-memory may give rise to a more powerful memory system. However (as we also mentioned in the same paragraph), given that hardware-efficiency and scalability is a hard requirement for machine learning models currently, such a design may not be straightforward for efficient training (but remains an open research question).
>
>
> We hope our responses above have successfully addressed the remaining concerns of the reviewer. Thank you again for all the excellent suggestions that help us improve the final version of the paper.
>
> ----
> **ADDITIONAL NOTE**: We will add one more set of experiments in the final revision. We conducted extra experiments to evaluate the Synchronous HQLT on **reinforcement learning in partially observable environments** involving retrieval, using the “passive visual match” task [Rebuttal Ref. 1]. This allows us to further validate our method on a domain outside of the language. The task is illustrated in Figure 2 of the corresponding paper [Rebuttal Ref. 1] (we are referring to this paper, because this year, the NeurIPS review system does not allow us to provide our own figures and plots during the rebuttal): In this task, an agent navigates in a 2D grid world (of size 7×11) delimited by impermeable walls. The agent can only observe the nearby pixels (5 × 5-grid centered on the agent). An episode in this task has three phases. During Phase 1 (whose
> duration is 15 time steps), the agent observes a color, randomly drawn from three choices, red, green or blue. In Phase 2 (750 steps), the agent is in a room with apples; collecting an apple yields a reward of 1. There are initially 10 apples, and they reappear every 20 steps; their positions are random. In Phase 3 (max. 15 steps), the agent is placed in a room with three colors; if the agent reaches the pixel with the color that matches the one provided in Phase 1, the episode ends successfully; it yields a reward of 100. Alternatively, Phase 3 terminates if the agent reaches a pixel with the wrong color or when the limit of 15 steps is reached (no reward is given in these cases).
>
> The episode/sequence length for this task is maximum 780. The attention window size for the baseline quadratic transformer is set to 780 such that it covers the full episode sequence. We set the KV-memory window size of HQLT to 64 (i.e., less than 10 times the transformer window size). We obtain the following result in terms of average return (over 3 seeds):
>
> |  Model                |  Average Return |
> |-------------|----------------|
> | DeltaNet    |   253.7 |
> | HQLT         |   311.4 |
> | Transformer  |  335.1 |
>
> In the final version, we will include the corresponding learning curves with 95% confidence interval, as is typically done to report RL experiments. We believe this will broaden the scope of the work and make it more appealing to the wider NeurIPS community.
>
> [Rebuttal Ref. 1] Ni et al. When do transformers shine in RL? Decoupling memory from credit assignment. NeurIPS 2023.

---

> > ### Comment · Reviewer_Ww5m · 2025-08-04
> >
> > I'd appreciate the authors for their rebuttal and the detailed clarifications.
> >
> > The response adequately addresses the main concerns raised in the initial review. Accordingly, I have decided to retain my original score.

---

### Official Review · Reviewer_Mb9d · 2025-07-07

**Clarity:** 3
**Significance:** 2
**Originality:** 2
**Rating:** 4
**Confidence:** 4

**Summary:**

This paper proposes a hybrid quadratic-linear architecture that combines two types of memory systems: key-value memory (used in transformers for precise recall but with quadratic complexity) and synaptic memory (used in linear transformers for efficient computation). The authors design and compare three hybrid models—Delayed-Streaming, Delayed-Chunk, and Synchronous—and show that the Synchronous hybrid best leverages the strengths of both memory types. Through experiments on language modeling, expressivity tasks, and in-context retrieval tasks, the paper demonstrates the hybrid model's enhanced performance and scalability.

**Questions:**

see weakness.

**Ethical Concerns:**

["NO or VERY MINOR ethics concerns only"]

**Limitations:**

see strength 3.

**Quality:**

2

**Strengths And Weaknesses:**

strengths

1. In-layer hybrid architecture series validated on linear attention with delta rule. Although incremental, it is still worthwhile to explore the combination of two very different memory mechanisms.

2. Experiments on different hybrid methods on multiple tasks and their detailed ablations.

3. The author's discussion of this work is valuable. The fixed-size memory of linear attention is difficult to accurately handle retrieval tasks, but the infinitely expanded kv memory is not necessarily optimal. Although the author did not provide a sufficiently novel solutions in the article, how to better combine the two types of memory is an important research topic.

weakness

1. The abstract and introduction lack a brief introduction to the method. The authors should have elaborated on the "Three blending methods"(i.e. various types of HQLT).

2. I thought that Delayed-Streaming HQLT and Delayed-Chunk HQLT just correspond to the serial and chunk parallel forms in linear attention, and the calculation results are the same. But from the authors' experiment, Delayed-Chunk HQLT is different from Delayed-Streaming HQLT in both calculation process and results, which confuses me. I hope the author can explain why Delayed-Chunk HQLT is a hybrid method independent of Delayed-Streaming HQLT.

3. How does the training efficiency of hybrid architectures compare to linear models and transformers? A supplementary experiment may answer my concerns.

4. The authors mentioned in the limitation that "Investigating such a mechanism is not straightforward in the current era of model development, which sets hardware-efficiency as a hard requirement." I value this point of view of the authors. Recently, Log-linear attention [1] has achieved a balance in exploring the utilization of intermediate-scale memory and computational efficiency. I wonder what the authors think of this work. (This is just an open question)

minors:
5. The word "synaptic memory" used in the abstract is confusing, as the authors does not provide any explanation in the paper. Either explain it in the text, or use FW memory instead, which may be easier to understand.
6. Some HQLTs are misspelled as HQTLs. Use ctrl+F to find them.

[1] Guo, et al. Log-Linear Attention.

---

> ### Author Rebuttal · Authors · 2025-07-30
>
> We thank the reviewer for their valuable feedback and overall positive comments.
> We sincerely appreciate your thoughtful and high-quality review. We believe that we have good explanations to resolve the reviewer’s remaining concerns. Please find our responses below.
>
> > 1. The abstract and introduction lack a brief introduction to the method. The authors should have elaborated on the "Three blending methods" (i.e. various types of HQLT).
>
> Thank you for this suggestion. We will amend the corresponding sentence in the abstract as follows:
> “We propose and compare three methods to blend these two systems into a single memory system, **differing in how and when input information is delivered to each system**, to leverage the strengths of both.”
> We will similarly improve the introduction (Line 35).
>
> > 2. I thought that Delayed-Streaming HQLT and Delayed-Chunk HQLT just correspond to the serial and chunk parallel forms in linear attention, and the calculation results are the same. But from the authors' experiment, Delayed-Chunk HQLT is different from Delayed-Streaming HQLT in both calculation process and results, which confuses me. I hope the author can explain why Delayed-Chunk HQLT is a hybrid method independent of Delayed-Streaming HQLT.
>
> The calculation results are different in Delayed-Streaming HQLT and Delayed-Chunk HQLT because the classic equivalence between recurrent-vs-chunk mode computation in linear attention does not hold when softmax is applied within the intra-chunk attention. To be more specific, the Delayed-Streaming variant has a sliding window attention with a stride of 1, i.e., **at every time step**, the oldest token from the KV-memory window is fed to the FW-memory, and a new token enters the KV-memory window; as a consequence, the memory content of FW-memory is updated at every time step, whereas in the Delayed-Chunk variant, the sequence is processed chunk-by-chunk, i.e., the content of FW-memory is only updated at the chunk-boundaries and remains constant while the system is processing the current chunk. Within a chunk, the content of KV-memory is updated at every time step by adding the new element in the chunk; its content is reset (to empty) at the chunk boundaries. Intuitively, Delayed-Streaming HQLT may be argued to be a better approach as the KV-memory always makes use of the full window size (except at the very beginning of the sequence), but we also included Delayed-Chunk which is not only a natural extension of chunk-wise parallel form of linear attention, but was used in prior work by Munkhdalai et al. [15].
>
> This is an important point we should have better clarified in the paper.  We will add a clarification sentence in Sec. 3.2 in the revision. Thank you very much for pointing this out.
>
> > 3. How does the training efficiency of hybrid architectures compare to linear models and transformers? A supplementary experiment may answer my concerns.
>
> The information in the supplementary material provides an answer to this question (Appendix A.1 Line 386): “Training of 340M models using 4 H100-80GB GPUs takes about 8 hours for the baseline transformer and 10 hours for DeltaNet and all the HQLT models with the window size of 64 tokens. For the 1.3B models, these numbers become 26 hours for the baseline transformer and DeltaNet, and 30 hours for all the HQLTs variants.”
>
> Given that the training for the 1.3B models uses 16B tokens, training throughputs are 170K tokens/second and 148K tokens/second for the baseline Transformer/DeltaNet and HQLT, respectively.
>
> > 4. The authors mentioned in the limitation that "Investigating such a mechanism is not straightforward in the current era of model development, which sets hardware-efficiency as a hard requirement." I value this point of view of the authors. Recently, Log-linear attention [1] has achieved a balance in exploring the utilization of intermediate-scale memory and computational efficiency. I wonder what the authors think of this work. (This is just an open question)
>
> Thank you for this open question for discussion. We believe it is a very interesting approach that nicely extends linear attention, while preserving the efficiency through an elegant parallel-form computation. That said, the fundamental memory mechanism (reading/writing primitives) of the proposed log-linear attention models still relies on those of the existing FW-memory models. In this sense, it is orthogonal to the comment we made in the Discussion, as our discussion was about future development to improve upon these memory primitives.
>
> > minors: 5. The word "synaptic memory" used in the abstract is confusing, as the authors does not provide any explanation in the paper. Either explain it in the text, or use FW memory instead, which may be easier to understand.  6. Some HQLTs are misspelled as HQTLs. Use ctrl+F to find them.
>
> Thank you very much for pointing these out. We will address these points in the final revision.
>
> We hope our responses above have successfully resolved the remaining concerns of the reviewer. If so, we would be grateful if the reviewer could consider raising the score. Thank you again for all the excellent suggestions that help us improve the final version of the paper.
>
> ---
> **ADDITIONAL NOTE**: We will add one more set of experiments in the final revision. We conducted extra experiments to evaluate the Synchronous HQLT on **reinforcement learning in partially observable environments** involving retrieval, using the “passive visual match” task [Rebuttal Ref. 1]. This allows us to further validate our method on a domain outside of the language. The task is illustrated in Figure 2 of the corresponding paper [Rebuttal Ref. 1] (we are referring to this paper, because this year, the NeurIPS review system does not allow us to provide our own figures and plots during the rebuttal): In this task, an agent navigates in a 2D grid world (of size 7×11) delimited by impermeable walls. The agent can only observe the nearby pixels (5 × 5-grid centered on the agent). An episode in this task has three phases. During Phase 1 (whose
> duration is 15 time steps), the agent observes a color, randomly drawn from three choices, red, green or blue. In Phase 2 (750 steps), the agent is in a room with apples; collecting an apple yields a reward of 1. There are initially 10 apples, and they reappear every 20 steps; their positions are random. In Phase 3 (max. 15 steps), the agent is placed in a room with three colors; if the agent reaches the pixel with the color that matches the one provided in Phase 1, the episode ends successfully; it yields a reward of 100. Alternatively, Phase 3 terminates if the agent reaches a pixel with the wrong color or when the limit of 15 steps is reached (no reward is given in these cases).
>
> The episode/sequence length for this task is maximum 780. The attention window size for the baseline quadratic transformer is set to 780 such that it covers the full episode sequence. We set the KV-memory window size of HQLT to 64 (i.e., less than 10 times the transformer window size). We obtain the following result in terms of average return (over 3 seeds):
>
> |  Model                |  Average Return |
> |-------------|----------------|
> | DeltaNet    |   253.7 |
> | HQLT         |   311.4 |
> | Transformer  |  335.1 |
>
> In the final version, we will include the corresponding learning curves with 95% confidence interval, as is typically done to report RL experiments. We believe this will broaden the scope of the work and make it more appealing to the wider NeurIPS community.
>
> [Rebuttal Ref. 1] Ni et al. When do transformers shine in RL? Decoupling memory from credit assignment. NeurIPS 2023.

---

> > ### Comment · Reviewer_Mb9d · 2025-08-03
> >
> > Thanks for your response. My most questions have been resolved. On the one hand, considering that Delayed-Streaming HQLT is actually a different approach from Delayed-Chunk HQLT, the authors could consider adding an illustration of Delayed-Chunk HQLT like Figure 1 to improve clarity in the future. On the other hand, I would like to follow up on W2. In Line 178, the authors state: “This approach (Delayed-Chunk HQLT) is naturally compatible with efficient training: we can apply flash-linear-attention for inter-chunk attention computation, and flash attention to the intra-chunk attention with softmax.” This raises the question: can Delayed-Streaming HQLT also support similarly chunk-wise efficient training? If so, this statement might be misleading, as it gives the impression that Delayed-Streaming HQLT must be trained in a strictly sequential (step-by-step) manner.

---

> > > ### Author Response · Authors · 2025-08-03
> > >
> > > Thank you very much for your response. We are pleased to hear that most of your questions have been resolved.
> > >
> > > Yes, we will consider adding a figure for the "Delayed-Chunk" variant in the revision.
> > >
> > > Regarding "Delayed-Streaming" vs. "Delayed-Chunk" HQLT, we agree that the current sentence you highlighted is misleading; because the FW-memory component of the "Delayed-Streaming" variant is still trained using a chunk-wise parallel form. We will fix this sentence in the revision to avoid this potential confusion. We thank the reviewer once again for pointing this out.

---

> > > > ### Comment · Reviewer_Mb9d · 2025-08-04
> > > >
> > > > Thank the author again for the response. Considering the completeness, importance, and novelty of this work, I am willing to maintain my score. I look forward to the final revision submitted by the author.

---

### Decision · Program_Chairs · 2025-09-17

**Decision:**

Accept (poster)

**Comment:**

Accept.

This paper provides a valuable and systematic study of how to best combine quadratic and linear attention mechanisms. Its key finding is that a "Synchronous" blending strategy preserves the expressivity of the linear component (DeltaNet), a crucial insight for designing future hybrid models.





While the practical performance gains on some standard benchmarks are modest, the authors' rebuttal significantly strengthened the paper. They introduced new experiments providing clear evidence that the hybrid approach offers genuine complementarity, outperforming a standard Transformer on retrieval tasks when controlling for window size. All reviewers converged on acceptance after the rebuttal , acknowledging the work's importance to the research community's growing interest in hybrid architectures.